

# The CAMS interim Reanalysis of Carbon Monoxide, Ozone and Aerosol for 2003–2015

J. Flemming[1], A. Benedetti[1], A. Inness[1], R. Engelen[1], L. Jones[1], V. Huijnen[2], S. Remy[3], M. Parrington[1], M. Suttie[1], A. Bozzo[1], V.-H. Peuch[1], D. Akritidis[4] and E. Katragkou[4]

[1] European Centre for Medium-Range Weather Forecasts, Reading, UK

[2] Royal Netherlands Meteorological Institute, De Bilt, The Netherlands

[3] Laboratoire de météorologie dynamique, UPMC/CNRS, Paris, France

[4] Department of Meteorology and Climatology, Aristotle University of Thessaloniki, School of Geology, Thessaloniki, Greece

Correspondence to: J. Flemming (Johannes.Flemming@ecmwf.int)



**Abstract**
A new global reanalysis data set of atmospheric composition (AC) for the period 2003–2015
has been produced by the Copernicus Atmosphere Monitoring Service (CAMS). Satellite
observations of total column (TC) carbon monoxide (CO) and aerosol optical depth (AOD) as
well as several TC and profile observation of ozone have been assimilated with the Integrated
Forecasting System for Composition (C-IFS) of the European Centre for Medium-Range
Weather Forecasting. Compared to the previous MACC reanalysis (MACCRA), the new
CAMS interim reanalysis (CAMSiRA) is of a coarser horizontal resolution of about 110 km
compared to 80 km but covers a longer period with the intent to be continued to present day.
This paper compares CAMSiRA against MACCRA and a control experiment (CR) without
assimilation of AC retrievals. CAMSiRA has smaller biases than CR with respect to
independent observations of CO, AOD and stratospheric ozone. However, ozone at the
surface could not be improved by the assimilation. The assimilation of AOD led to a global
reduction of sea salt and desert dust as well as an exaggerated increase in sulphate. Compared
to MACCRA, CAMSiRA had smaller biases for AOD, surface CO and TC ozone as well as
for upper stratospheric and tropospheric ozone. Finally, the temporal consistency of
CAMSiRA was clearly better than the one of MACCRA. This was achieved by using a
revised emission data set as well as by applying a careful selection and bias-correction of the
assimilated retrievals. CAMSiRA is therefore better suited than MACCRA for the study of
inter-annual variability than MACCRA as demonstrated for trends in surface CO.





## 1   Introduction

Exploiting the multitude of satellite observations of atmospheric composition (AC) is a key
objective of the Copernicus Atmosphere Monitoring Service (CAMS). For its global
component CAMS uses the four-dimensional variational (4D-VAR) data assimilation
technique to combine satellite observations with chemistry-aerosol modelling to obtain a
gridded continuous representation (analysis) of the mass mixing ratios of atmospheric trace
gases and aerosols.
The global CAMS system is built on the heritage of the EU-funded GEMS (Hollingsworth et
al., 2008) and series of MACC projects at the European Centre for Medium-Range Weather
Forecasts (ECMWF). During these projects the Integrated Forecasting System (IFS) of
ECMWF was extended by modules for atmospheric chemistry, aerosols and greenhouse gases
in such a way that the 4D-VAR data assimilation system, which had been developed for the
analysis of the meteorological fields, could be used for the assimilation of AC retrievals.
Assimilating satellite AC retrievals into an AC model has advantages to the sole use of the
AC retrievals because of their specific limitations. First, only a small subset of the trace gases
or only total aerosol is directly observable with sufficient accuracy. Second, AC satellite
retrievals have incomplete horizontal coverage because of the orbital cycle, viewing
geometry, the presence of clouds and other factors such as surface properties. Third, the
vertical distribution of the trace species can often not or only rather coarsely be retrieved from
the satellite observations, while the measurement sensitivity towards the surface is generally
low.
The AC analyses are used to (i) initialise AC model forecasts and (ii) for the retrospective
analysis (reanalysis) of AC for air quality and climate studies. The reanalysis of the
meteorological fields has been an important activity at ECMWF (ERA-40, Uppala et al.,
2005, ERA interim Dee et al., 2011) and other meteorological centres such as NCEP (CFSR,
Saha et al., 2010, JMA (JRA-55, JRA-25, Onogi et al., 2007) and NASA/DAO (MERRA,
Rienecker, et al., 2011). An important application of these reanalysis data sets is the
estimation of the inter-annual variability and the trends of climate variables over the last
decades up to the present day. The complete spatial and temporal coverage makes the trend
analysis of reanalyses more robust and universal than the trend analysis of individual
observing systems. However, constructing a data set which is suited for this purpose is a
complex task because of the developing and changing observing system, which can introduce





73 spurious trends and sudden shifts in the reanalysis data record. Careful quality control of the

74 assimilated observations and techniques (e.g. Dee et al., 2004) to address inter-instrument

75 biases are applied to mitigate this problem.

76 Most meteorological reanalyses contain stratospheric ozone but other traces gases, apart from

77 water vapour, are not included. In the last decade chemical and aerosol data assimilation has

78 matured (Bocquet et al., 2015) and dedicated reanalysis data sets for AC have emerged. The

79 Multi-Sensor-Reanalysis of total ozone (van der A et al., 2015) for 1970–2012 used ground

80 based Brewer observations to inter-calibrate satellite retrievals. The MERRAero reanalysis

81 (2002–present, http://gmao.gsfc.nasa.gov/reanalysis/merra/MERRAero/) assimilated AOD

82 retrievals from the two Moderate Resolution Imaging Spectroradiometer (MODIS) instruments

83 in the GOCART aerosol module of the GEOS-5 model system using the meteorological

84 variables of the MERRA meteorological analysis. Its next version, the MERRA2 reanalysis, is

85 a joint meteorological and aerosol reanalysis covering the period from 1979 to present.

86 Miyazaki et al. (2015) put together a tropospheric chemistry reanalysis using a Kalman filter

87 approach for the years 2005–2012. They use the CHASER Chemical transport model (CTM)

88 to assimilate retrievals of tropospheric ozone and CO profiles, $NO_2$ tropospheric columns and

89 $HNO_3$ stratospheric columns. Their approach tackles two specific challenges of AC data

90 assimilation. First, they not only correct atmospheric concentrations but also alter the surface

91 emissions which control the tracer distributions to a large extent. Second, the Kalman filter

92 develops co-variances of the errors between observed and un-observed species, which are used

93 to correct un-observed species based on the observations increments.

94 The MACC reanalysis (MACCRA) of reactive gases (Inness et al., 2013) and aerosols for the

95 period 2003–2012 is an AC reanalysis that covers tropospheric and stratospheric reactive gases

96 and aerosols as well as the meteorological fields in one consistent data set. MACCRA has

97 proved to be a realistic data set as shown in several evaluation studies for reactive gases

98 (Elguindi et al., 2010, Inness et al., 2013, Katragkou et al., 2015 and Gaudel et al., 2015) and

99 aerosols (Cesnulyte et al., 2014 and Cuevas et al., 2015, ). MACCRA is widely used, for

100 example, as boundary condition for regional models (Schere et al., 2012, Im et al., 2014,

101 Giordano et al., 2015), to construct trace gas climatologies for the IFS radiation schemes

102 (Bechtold et al., 2009), to estimate aerosol radiative forcing (Bellouin et al., 2013), as input to

103 solar radiation schemes for solar energy applications and to report the current state of aerosol

104 and CO as part of the climate system (Benedetti et al., 2014., Flemming and Inness, 2014).



CAMS is committed to produce a comprehensive high-resolution AC reanalysis in the next
years. The CAMS interim Reanalysis (CAMSiRA) presented here has an interim status
between MACCRA and this planned analysis data set. It was produced at a lower horizontal
resolution (110 km) than the resolution of MACCRA (80 km), and the number of archived
AC fields was limited to selected key species only.
The reasons for producing CAMSiRA before the more comprehensive reanalysis are as
follows: The MACCRA for reactive gases was produced using a coupled system consisting of
the IFS and the MOZART-3 (Kinnison et al., 2007) chemical transport model (CTM) as
described in Flemming et al. (2009). This coupled system was replaced by the much more
computationally efficient on-line coupled model C-IFS (Flemming et al., 2015), which uses
the chemical mechanism CB05 of the TM5 CTM (Huijnen et al., 2010). With the
discontinuation of the coupled system it was not possible to extend the MACC reanalysis to
the present day. For the AC monitoring service of CAMS it is however important to be able to
compare the present conditions with previous years in a consistent way. Another motivation
for producing CAMSiRA was that the aerosol module used for the MACCRA had undergone
upgrades (Morcrette et al., 2011) in recent years. Finally, MACCRA suffered from small but
noticeable shifts because of changes in the assimilated observations, the emission data and the
bias correction approach. These spurious shifts undermine the usefulness of the MACCRA for
the reliable estimation of trends. The lessons learnt from the evaluation of CAMSiRA will
feed into the setup of the planned CAMS reanalysis.
Reanalyses of AC are generally less well-constrained by observations than meteorological
reanalyses because of the aforementioned limitations of the AC observations and because of
the strong impact of the emission, which are in many cases not constrained by observations. It
is therefore good scientific practise to investigate the impact of the AC assimilation by
comparing the AC reanalysis to a control experiment that did not assimilate AC observations.
The control run (CR) to CAMSiRA was carried out using the same emission data as well as
the meteorological fields produced by CAMSiRA.
The purpose of this paper is firstly to document the model system, the emissions and the
assimilated observations used to produce CAMSiRA, and to highlight the differences to the
setup of the MACCRA. As the emissions are an important driver for variability of AC, a
presentation of the totals and the inter-annual variability of the emission data used in
CAMSiRA and CR is given in a supplement to the paper.



In the remainder of the paper, CO, aerosol as well as tropospheric and stratospheric ozone of
CAMSiRA, CR and MACCRA are inter-compared and evaluated with independent
observations in a separate section for each species. The comparison of CAMSiRA with
MACCRA has the purpose to report progress and issues of CAMSiRA for potential users of
the data sets. The comparison of CAMSiRA with CR shows the impact of the data
assimilation and is helpful to better understand deficiencies of the C-IFS model and its input
data.
Each section starts with a discussion of the spatial differences of CAMSiRA, CR and
MACCRA of the considered species. Next, the temporal variability is investigated using time
series of monthly mean values averaged over selected regions. We present global burdens and
discuss changes in the speciation of the aerosol fields introduced by the assimilation. Finally,
the three data sets are compared against independent observations, which were not used in the
assimilation. A summary and recommendations for future AC reanalysis will be given in the
last section.

## 2  Description of CAMSiRA setup
### 2.1  Overview
CAMSiRA is a data set of 6 hourly reanalyses of AC for the period 2003–2015. A 3 hourly data
set consistent with the AC analysis is available from forecasts linking the analyses. The
horizontal resolution is about 110 km on a reduced Gaussian grid (T159) and the vertical
discretisation uses 60 levels from the surface to a model top of 0.1 hPa. Total columns of CO
(TC CO) of the Measurements Of Pollution In The Troposphere (MOPITT) instrument, MODIS
AOD and several ozone TC and stratospheric profile retrievals (see Table 2) were assimilated
together with meteorological in-situ and satellite observations.
The description of MACCRA for reactive gases can be found in Inness et al. (2013). Important
commonalities and differences between the two AC reanalyses are given in Table 1.
The control run is a forward simulation of C-IFS in monthly segments. The meteorological
simulation is relaxed using the approach by Jung et al. (2008) to the meteorological reanalysis
produced by the CAMSiRA. The emission input fields are the same as used for CAMSiRA.
### 2.2  C-IFS model
C-IFS is documented and evaluated in Flemming et al. (2015). C-IFS applies the chemical
mechanism CB05, which describes tropospheric chemistry with 55 species and 126 reactions.
Stratospheric ozone chemistry in C-IFS is parameterized by the "Cariolle-scheme" (Cariolle
and Dèquè, 1986 and Cariolle and Teyssèdre, 2007). Chemical tendencies for stratospheric and
tropospheric ozone are merged at an empirical interface of the diagnosed tropopause height in
C- IFS. C-IFS benefits from the detailed cloud and precipitation physics of the IFS for the
calculation of wet deposition and lightning NO emission. Wet deposition modelling for the
chemical species is based on Jacob (2000) and accounts for the sub-grid scale distribution of
clouds and precipitation. Dry deposition is modelled using pre-calculated monthly-mean dry
deposition velocities following Wesely (1989) with a superimposed diurnal cycle. Surface
emissions and dry deposition fluxes are applied as surface boundary conditions of the diffusion
scheme. Lightning emissions of NO were calculated based on convective precipitation (Meijer
et al., 2001).
The aerosol module (Morcrette et al., 2009) is a bulk/bin scheme simulating desert dust, sea
salt at 80% relative humidity (RH), hydrophilic and hydrophobic organic carbon and black





carbon as well as sulphate aerosol based on the LMDZ aerosol model (Reddy et al., 2005). Sea
salt and desert dust are represented in 3 size-bins. The radii ranges of the dust bins are 0.030–
0.55, 0.55–0.9 and 0.9–20 µm (DD1, DD2, and DD3) and of the sea salt at 80% RH bins 0.03–
0.5, 0.5–5 and 5–20 µm (SS1, SS2, and SS3). There is no consideration of the aerosol growth,
which would transfer aerosol mass from one size bin to another. Hygroscopic growth of
hydrophilic species is taken into account in the computation of the aerosol optical properties
only. Following the emission release, the aerosol species are subject to wet and dry deposition
and the largest size bins of sea salt and dust also to sedimentation. The chemical source of
sulphate is modelled by climatological conversion rates using a $SO_2$ tracer, which is
independent of the $SO_2$ simulated in CB05. The $SO_2$ tracer is driven by prescribed $SO_2$ and
DMS emissions. Its loss is simulated by wet and dry deposition as well as the climatological
chemical conversion to $SO_4$.
The aerosol and chemistry modules used to simulate source and sink terms are not coupled.
Also, wet and dry deposition are modelled with different parameterisations but with the same
meteorological input such as precipitations fields. Aerosol and chemistry have in common that
they are advected and vertically distributed by diffusion and convection in the same way. A
proportional mass fixer as described in Diamantakis and Flemming (2014) is applied for all
tracers in C-IFS.

## 2.3   Emission data sets

This section only references the origin of the emission data. The emitted totals and the linear
trends of the anthropogenic, biomass burning and natural emissions as well as the modelled
desert dust and sea salt emissions used in CAMSiRA and CR are presented in a supplement.
The anthropogenic surface emissions for the chemical species were taken from the MACCity
inventory (Granier et al., 2011), which covers the period 1960–2010. MACCity emissions are
based on the ACCMIP (Lamarque et al., 2013) inventory but have improved seasonal
variability. The changes from 2000–2005 and for 2010 are obtained using the representative
concentration pathways (RCP) scenarios version 8.5. For the production of CAMSiRA the
MACCity data set was extended to 2015 by also applying the RCP 8.5 scenario. The
anthropogenic CO emissions were increased following Stein et al. (2014). Time series of the
anthropogenic CO emissions for Europe, North America, East Asia (see Table 3) and the globe
are shown in Figure S2 of the supplement.



The anthropogenic emissions of organic matter, black carbon and aerosol precursor $SO_2$ are
retrieved from AEROCOM data base, which is compiled using EDGAR and SPEW data
(Dentener et al., 2006). In contrast to the anthropogenic gas emissions, the aerosol
anthropogenic emissions did not account for trends but only for the seasonal cycle.
The biogenic emissions for the chemical species were simulated off-line by the MEGAN2.1
model (Guenther et al., 2006) for the 2000–2010 period (MEGAN-MACC, Sindelarova et al.,
2014). For the remaining years 2011–2015 a climatology of the MEGAN-MACC data was put
together. Natural emissions from soils and oceans for $NO_2$, DMS and $SO_2$ were taken from
POET database for 2000 (Granier et al., 2005; Olivier et al., 2003).
Daily biomass burning emissions for reactive gases and aerosols were produced by the Global
Fire Assimilation System (GFAS) version 1.2, which is based on satellite retrievals of fire
radiative power (Kaiser et al., 2012). This is an important difference with respect to the
MACCRA, which used an early version of the GFED 3.1 data from 2003 until the end 2008
and daily GFAS v1.0 data from 2009 to 2012. The GFED 3.1 is on average 20% lower than
GFAS v1.2 (Inness et al., 2013). Time series of the biomass burning CO emissions for Tropical
Africa, South America and Maritime South East Asia (see Table 3) and the globe are shown in
Figure S3 of the supplement.

## 2.4   C-IFS data assimilation

C-IFS uses an incremental 4D-VAR algorithm (Courtier et al., 1994), which minimizes a cost
function for selected control variables to combine the model and the observations in order to
obtain the best possible representations of the atmospheric fields. The mass mixing ratios of $O_3$,
CO and total aerosol are incorporated into the ECMWF variational analysis as additional
control variables and are minimized together with the meteorological control variables. The
assimilation of satellite retrieval of the chemical species and total aerosol optical depth is
documented in Inness et al. (2015) and Benedetti et al. (2009). The assimilation of aerosol
differs from the assimilation of CO and ozone because only the total aerosol mass can be
constrained by the observations and information about the speciation must be obtained from the
model.
The assimilation of AOD retrievals uses an observation operator that translates the aerosol mass
mixing ratios and humidity fields of C-IFS to the respective AOD (550 nm) values using pre-
computed optical properties. Total aerosol mass mixing ratio is included in the 4D-VAR cost





function and the analysis increments are repartitioned into the individual aerosol components
according to their fractional contribution to the total aerosol mass. This is an approximation
which is assumed to be only valid over the 12 hour of the assimilation window. In reality, the
relative fraction of the aerosol components is not conserved during the whole assimilation
procedure because of differences in the efficiency of the removal processes. Aerosol
components with a longer atmospheric lifetime will retain relatively longer the change imposed
by the increments and may thereby change the relative contributions.
The background error statistics for the chemical species and for total aerosol are univariate in
order to minimize the feedback effects of the chemical fields on the meteorological variables.
Correlations between the background errors of different chemical species are also not accounted
for (Inness et al., 2015).
In the ECMWF data assimilation system the background error covariance matrix is given in a
wavelet formulation (Fisher, 2004, 2006). This allows both spatial and spectral variations of
the horizontal and vertical background error covariances. The background errors for AC are
constant in time.
The background errors for ozone are the same as the ones used for MACCRA (Inness et al.,
2013). Only the vertical correlations of the ozone background errors have been modified and
restricted to ± 5 levels around a model level, to avoid correlations between the lower
troposphere and upper tropospheric and stratospheric levels that affected near-surface ozone
adversely. The background errors of total aerosol for both MACCRA and CAMSiRA were
calculated using the method described in Benedetti and Fisher (2008). The aerosol background
errors for CAMSiRA were updated using a more recent C-IFS model version. The background
errors for CO are newly calculated for the CAMSiRA from an ensemble of C-IFS forecast runs
(Inness et al., 2015).
**2.5   Assimilated observations**
Table 2 shows the AC composition data sets for CO, ozone and AOD that were assimilated in
CAMSiRA. The time line of the assimilation for the different retrievals is shown in Figure 1.
CO is assimilated from MOPITT V5 TIR only whereas the MACCRA assimilated the V4 TIR
product and additionally IASI TC CO retrievals after April 2008. The biases between the
retrievals (George et al., 2015) of the two instruments in mid and higher latitudes could not be
reconciled with the variational bias correction and led to a discontinuity in the time series of



CO in MACCRA, which consequently could not be used for trend analyses (see Figure 4
below). It was therefore decided to only use the MOPITT V5 CO data set in CAMSiRA because
it covers the whole period from 2003–2015. The MOPITT V5 product has better long term
stability and a smaller SH bias than V4 (Deeter et al., 2013). V4 suffered from a positive
temporal bias drift and a positive bias in SH.
An additional ozone data set in CAMSiRA were the Michelson Interferometer for Passive
Atmospheric Sounding (MIPAS) ozone profiles, which were assimilated from 2005 until the
end of the ENVISAT mission in April 2012. After the end of 2012 the version of the assimilated
Microwave Limb Sounder (MLS) data set changed from V2 to V3.4. Information about the
differences        between        the        two        versions        can        be        found        in
https://mls.jpl.nasa.gov/data/v3_data_quality_document.pdf
Averaging kernels were used for the calculation of the model's first-guess fields in the
observation operators for the MOPITT data.
The AC satellite retrievals were thinned to a horizontal resolution of 1° x 1° by randomly
selecting an observation in the grid box to avoid oversampling and correlated observation
errors. Variational quality control (Andersson and Järvinen, 1999) and background quality
checks were applied. Only 'good' data were used in the analysis and data flagged as 'bad' by
the data providers were discarded.
Variational bias correction (Dee, 2004, McNally et al., 2006, Auligné et al., 2007, Dee and
Uppala, 2009) was applied to the MODIS AOD data, as well as to ozone column data from the
Ozone Monitoring Instrument (OMI), the SCanning Imaging Absorption spectroMeter for
Atmospheric CHartographY (SCIAMACHY) and the Global Ozone Monitoring Experiment 2
(GOME-2). The partial column of the Solar Backscatter Ultraviolet Radiometer 2 (SBUV/2),
MLS and MIPAS were used to anchor the bias correction. Experience from the MACC
reanalysis had shown that it was important to have an anchor for the bias correction to avoid
drifts in the fields (Inness et al., 2013).





## 3 Carbon monoxide


Global CTMs tend to underestimate the observed CO values (Shindell et al., 2006) but data
assimilation (Inness et al., 2013 and 2015, Miyazaki et al., 2015, Gaubert et al., 2016) of satellite
retrieval is able to successfully reduce the biases of the simulated CO fields. The correct
representation of vertical CO profiles by the assimilation remains a challenge (Gaudel et al.,
2015). An important next step will be the correct representation of the global CO trends by
means of CO reanalyses such as CAMSiRA.

### 3.1 Spatial patterns of total column CO


Figure 2 shows the seasonal mean of TC CO over the period 2003–2015 of CAMSiRA and the
differences with CR and MACCRA (2003–2012). Overall, the assimilation of TC CO in
CAMSiRA led to an increase in the northern hemisphere (NH) and a decrease in the Southern
hemisphere (SH) and most of the tropics. CAMSiRA was about 2–5% higher than CR in NH
and up to 20% lower in the SH. The reduction was especially large in the tropical and sub-
tropical outflow regions of the biomass burning regions in South America, Central Africa and
Maritime South East Asia. The largest reduction in these regions occurred in DJF. The largest
negative bias of CR with respect to CAMSiRA occurred over NH in December–February (DJF)
and March–May (MAM). Overall the zonal patterns of the biases throughout all seasons were
rather uniform indicating an underestimation of the hemispheric CO gradient in CR and could
point to deficiencies in the simulation of the global chemical loss and production of CO as well
as problems with the large scale transport. Biases in the amount of the emissions seem to play
a smaller role for the problem with the hemispheric gradient.
However, more emission related differences occurred in September–November (SON) and to
a smaller extent in June–August (JJA), when CR had (i) higher values in the biomass burning
regions and the respective outflow regions in Central Africa, Maritime South East Asia and
South America and (ii) lower values in the outflow regions of the emissions in North America
and East Asia in the Eastern and Western Northern Pacific. This suggests that GFAS biomass
burning emissions were too high whereas the anthropogenic emissions in North America and
East Asia were too low. On the other hand, CR had higher values than CAMSiRA in South
Asia, which indicates that the anthropogenic emissions are too high in India.
Compared to MACCRA, CAMSiRA was up to 10% higher in the Northern high latitudes and
up to 20% higher above the tropical biomass burning regions and above the parts of East



Asia.The differences over the biomass burning regions can be attributed to the different biomass
burning emissions data sets (see section 2.3). Over the oceans in NH and the tropics, apart from
biomass burning outflow regions, CAMSiRA CO is slightly lower (3%) than MACCRA. The
differences in the NH high latitudes are mainly caused by the reduction in MACCRA CO in
this region introduced by the assimilation of IASI CO retrieval after 2008 (see also Figure 4
below).
Figure 3 shows the average zonal mean cross section of the average CO mass mixing ratio of
CAMSiRA and the relative difference to CR and MACCRA. The overestimation of CR in the
tropics and SH extratropics was found throughout the troposphere. It was most pronounced in
relative terms at about 500 hPa. Stratospheric CO in CAMSiRA was much lower than in
MACCRA. This might be an improvement as Gaudel et al. (2015) report an overestimation in
the MACCRA over this region. In the upper troposphere CAMSiRA had higher CO than
MACCRA most notably in the tropics and SH where values are up to 40% higher. CO was
lower in the mid and lower troposphere in SH and higher in NH. These differences in the
vertical distribution might be caused by (i) a more consistent modelling approach of the
stratosphere-troposphere exchange with the on-line coupled C-IFS, (ii) the fact that C-IFS
CB05 has a very different chemistry treatment compared to MOZART and (iii) updated
background error statistics for CO (see Table 1).
**3.2   Inter-annual variability of CO burden**
Figure 4 shows time series of the monthly mean CO burden from CAMSiRA, MACCRA and
CR for selected areas (see Table 3). Then modelled global CO burden (CR) was reduced by the
assimilation by about 3% at the start and by about 7% at the end of the period. CAMSiRA
showed a stepwise decrease of the global CO burden from 2008 and 2009 which corresponds
to a significant negative linear trend of -0.86%/yr over the whole period. This figure is in good
agreement with the results of Worden et al. (2013) who estimates trends of -1% per year for
both the globe and NH over the last decade by studying different satellite-based instruments.
CR also showed the largest decrease in the period from 2007–2009 but the CO burden increased
slightly after that period. The resulting linear trend of CR was still negative (-0.36%/yr) but less
strong than the trend of CAMSiRA.
The higher global CO burdens of CR with respect to CAMSiRA originated mainly from the
tropics and the SH mid-latitudes, which are strongly influenced by biomass burning emissions



in tropical Africa and South America. CO was reduced by the assimilation in CAMSiRA
especially after the start of the biomass burning season. The reduction of the biomass burning
emissions of -7.4%/yr (see supplement Table S1) over South America led to a significant
negative trend of the CO burden of -1.23%/yr in CAMSiRA and -0.83%/yr in CR over that
region. The overestimation of CR with respect to CAMSiRA increased slightly during the
period.
2015 was an exceptional year because the global CO burden reached the highest values in the
whole period for both CAMSiRA and CR despite the overall decadal negative trend. The
increase was caused by exceptionally high biomass burning emissions in Indonesia because of
El Niño related dry conditions. The El Niño controlled inter-annual variability of CO over
Maritime South East Asia was reproduced in a very similar way in CAMSiRA and CR but the
assimilation reduced the burden by about 1 Tg (10%).
In the regions of high anthropogenic emissions the temporal variability at a monthly scale was
very similar between CR and CAMSiRA. Both in North America and Europe CR
underestimated the CO maximum of CAMSiRA in early spring by less than 5% up to the year
2010 but the biases almost disappeared in later years. This means that the negative total CO
trend in these regions was larger in CAMSiRA, which contains the MOPITT observations, than
in CR. It could indicate that the anthropogenic emissions were biased low at the beginning of
the period but less so towards the end. Over East Asia the difference between CR and
CAMSiRA was generally very small indicating a high degree of realism of the emissions in the
area. A further explanation for this agreement is the fact that this area covers both the
underestimation of CAMSiRA by CR in NH mid-latitudes and the overestimation in the tropics.
Both CAMSiRA and CR had a negative but not a significant tend over East Asia.
Stroden et al. (2016) also find good agreement between MOPITT-based and modelled negative
trends for the 2000-2010 period of total column CO over Europe and North America but
disagreement in the in the sign of the trend over Eastern China, where their model, using
MACCity emissions, simulates a positive trend but MOPITT has a negative trend. Over Eastern
China also CR (2003-2015) had a small positive linear trends whereas CAMSiRA had a
negative trend but both trends were not statistically significant. The positive trend over Eastern
China in CR was mainly driven by directly emitted CO at the surface. Owing to the hemispheric
influence, the CO trend in CR became negative in the middle troposphere, where the MOPITT
sensitivity to CO is highest.



In the Arctic, which is influenced by the long-range transport from North America, Europe and
Asia (Emmons et al., 2015), no MOPITT observations were assimilated (see Table 2). Also in
this region the variability of the CR and CAMSiRA CO burden matched well but the bias was
much reduced after 2012.
The time series of the global CO burden of CAMSiRA and MACCRA agree better than
CAMSiRA and CR. The global burden of MACCRA is slightly lower than in CAMSiRA (1%)
until 2010 but starts to exceed CAMSiRA in 2011 and 2012. Hence, larger differences occur at
the beginning and end of the MACCRA period.
The CO burden of MACCRA above the biomass burning regions of South America and
Tropical Africa was lower than CAMSiRA for the period 2003–2010. This is most likely
because of the use of the GFED biomass burning emissions until 2008, which are on average
20% lower than GFAS, which was used for CAMSiRA . In the years 2011–2012 MACCRA
had higher values, which even led to a reversal in the sign of the trend over the two regions in
MACCRA in comparison to CAMSiRA. MACCRA and CAMSiRA agreed well above the
anthropogenic source regions. Only from 2008 onwards MACCRA was slightly lower which
led to enhanced negative trends.
Over the Arctic, CAMSiRA is higher from 2008 whereas MACCRA was higher at the start.
This is consistent with the respective trends over Europe and North America. All data sets
showed a step-like reduction the CO burden at mid-2008 but it was most pronounced in
MACCRA.

### 3.3  Evaluation with MOZAIC/IAGOS aircraft CO observations

Measurements of OZone, water vapour, carbon monoxide and nitrogen oxides by in-service
AIrbus aircraft (MOZAIC) and In-service Aircraft for a Global Observing System (IAGOS) are
subsequent programmes of AC observations mounted on commercial aircraft. The MOZAIC
CO data have an accuracy of $\pm$ 5 ppbv, a precision of $\pm$ 5%, and a detection limit of 10 ppbv
(Nédélec et al., 2003). De Laat et al. (2014) compare MOZAIC/IAGOS profile with the
MOPITT v5 NIR retrievals, which were assimilated in CAMSiRA. They find good agreement
and no drift of the biases of the two data sets in their study period 2002–2010.



We use the CO profiles obtained during take-off and landing to evaluate the CO fields averaged
over airports in different regions from 2003–2012. The number of MOZAIC/IAGOS CO
profiles fluctuated considerably over the years. They have decreased from 2003–2014 by about
50% and certain airports had many more observations than others. Since the aircraft used in
MOZAIC were based in Frankfurt, the majority of the CO profiles were observed at this airport.
Therefore the observations from Frankfurt dominate the European mean values. Observations
from Tokyo and other Japanese cities were the largest contribution to the mean over East Asia.
Atlanta, Toronto and Vancouver had the largest number of observation in the North American
domain. Windhoek had by far the largest number of observations in Tropical Africa and Caracas
in South America. The mean of Maritime South East Asia sea salt is mainly calculated from
observations over Jakarta and Kuala Lumpur in 2005, 2006, and 2012 with an unbalanced
coverage of the difference months.
Profiles of the mean relative bias of CAMSiRA, MACCRA and CR against MOZAIC/IAGOS
CO observations for different regions (see Table 3) averaged over the period 2003–2012 are
shown in Figure 5. We discuss here only the annual biases since the seasonal relative biases did
not differ to a large extent from the annual relative biases.
All three data sets underestimated the observed CO values throughout the troposphere in
Europe, North America and East-Asia. At the surface and the lower PBL up to 900 hPa, i.e.
where the highest CO concentrations are observed, CAMSiRA and CR had a relative biases of
about -10% in Europe and North America and up to -20% in East Asia, whereas MACCRA had
larger relative biases of -20 –-30% at this level and the largest biases occurred in DJF. On the
other hand, MACCRA had smaller biases than CAMSiRA and CR in the middle and upper
troposphere. The smaller biases of MACCRA may be caused by the more realistic simulation
of the chemical CO production by the MOZART chemical mechanism as well as by the change
in the CO background error statistic. The assimilation of MOPITT in CAMSiRA reduced the
biases relative to CR in the troposphere over Europe and North America but had only little
effect at the surface. Over East Asia the assimilation did not lead to changes between CR and
CAMSiRA.
Whereas CR had the largest underestimation in NH it was generally higher than CAMSiRA and
MACCRA in the tropics. This led to better agreement with the MOZAIC observation in South
America and Tropical Africa but also to an overestimation of 20–30% in Maritime South East
Asia. The limited number of observations in that region makes this result less robust. MACCRA




and CAMSiRA showed little differences over South America and Tropical Africa. The 10%
negative bias of MACCRA and CAMSiRA in Tropical Africa is consistent with the 10%
underestimation of MOPITT v5 against MOZAIC/IAGOS over Windhoek reported by de Laat
et al. (2014, their Figure 3). Over MSEA below 700hPa CAMSiRA and MACCRA
overestimated CO whereas MACCRA underestimated the observations. This could be the
consequence of the different fire emissions and the different chemistry schemes but the limited
number of available profiles makes this result less representative.

### 3.4   Evaluation with NOAA GMD surface observations

NOAA Global Monitoring Division (GMD) network of flask CO surface observations (Novelli
and Masarie, 2010) has a good global coverage, which also includes the high latitudes of SH
and NH, to observe the background concentrations. The tropical stations represent the maritime
background because they are mainly located on islands in the tropical oceans. The station
density is higher in North America and Europe. The uncertainty of the NOAA/GMD CO
observations is estimated to be 1–3 ppm (Novelli et al., 2003).
We calculated the mean and linear trend at each station for the period 2003–2014 or 2003-2012
(MACCRA). The overall bias averaged over all stations of CAMSiRA and CR was 3.0 ppb for
the whole period but CAMSiRA had a slighter lower RMSE (13 ppb) than CR (15 ppb). For
the 2003–2012 period MACCRA had a bias of 6 ppb whereas CAMSiRA and CR had a bias of
3.1 and 3.9 ppb respectively.
Figure 6 shows the zonal means of the observed averages and the corresponding model values
at station location as well as the median of the estimated linear trend from the observations and
the model results. The graphs were constructed by calculating the mean concentrations and
median trends of all stations in 15° wide latitude bins. The errors bars indicate the range of the
observed values in the latitude bin.
In the SH high and mid-latitudes the typical observed annual mean surface concentration was
50 ppbv. The background levels started to rise in the SH extra tropics and reached a maximum
of 145 ppbv in the NH mid- latitudes. The values then decreased to about 130 ppb in the Arctic.
The general structure of the zonal variation was well represented by all data sets. CR
overestimated the SH mid and high values by 15 ppb whereas CAMSiRA and MACCRA had
a bias of 7 ppb. In the tropics CAMSiRA had slightly lower (3 ppb) values than the observations
whereas MACCRA and CR overestimated by about 5 ppb. CAMSiRA had the highest values





of all three data sets in the NH mid-latitudes but still underestimated the mean of the
observations by 7 ppb. However the observed means at the station locations in this latitude band
varied in a range of about 100 ppb. CR had a slightly larger underestimation than CAMSiRA.
MACCRA underestimated the observations by more than 20 ppb in the mid and high latitudes.
The reduction towards the NH high latitudes in CR and CAMSiRA was similar to the
observations.
The observations in the SH showed essentially no linear trend in the 2003–2014 period. Starting
in the tropics a negative linear trend gradually occurred which reached values of about -2.2
ppb/yr in the NH mid- and high latitudes. CAMSiRA and CR had a small but still significant
(95% confidence level) negative trend in SH of -0.3 and -0.5 ppb/yr respectively. The negative
trends of CAMSiRA and CR started to become more pronounced from 20°S onwards. The trend
in CAMSiRA was generally stronger than the trend in CR. This meant a better fit with the
observed trends in the tropics for CR and a better fit in the NH mid- and high latitudes for
CAMSiRA. In this region the median of the trends was -2.1ppb/yr for CAMSiRA and -2.0
ppb/yr for CR. While the trends of CAMSiRA and CR agreed reasonably well with the
observations, MACCRA suffered from unrealistically strong negative trends in the mid- and
high latitudes of both hemispheres. This negative trend in MACCRA was caused by the
reduction in the values related to assimilation of IASI data from 2008 onwards (Inness et al.,

506   2013).



## 509   4  Aerosols

In contrast to the assimilation of individual chemical gases, the assimilation of AOD
observations is "underdetermined" because different combinations of the aerosol components
can led to the same extinction, i.e. AOD value. A further complicating factor is that each aerosol
component has different optical properties, which depend on relative humidity for the
hydrophilic components such as sea salt, sulphate and organic matter. The correction of the
speciation of the assimilated aerosol mass mixing ratio fields is therefore a big challenge despite
good success in reproducing independent AOD observations with the aerosol analysis.

### 4.1 Global aerosol burden, speciation and AOD


In this section the global averages of burdens and AOD are presented. Spatial patterns of AOD
will be discussed in section 4.2. Global area-weighted averages of AOD at 550nm and the total
global burden in Tg for the different aerosol components are shown in Figure 7. The figure also
shows the median of the global AOD average and burdens simulated by the models of the
AeroCom inter-comparison study (Kinne et al., 2006 and Textor et al., 2006). CR had the
highest total global average aerosol burden of 46 Tg compared to MACCRA and CAMSiRA,
which had both 33 Tg. This number was very similar to the AeroCom median of 29 Tg.
The global sea salt burden was about twice as high in CR (15.1 Tg) than in CAMSiRA (8.3
Tg), and it was 16.1 Tg for MACCRA. In comparison, the median of the sea salt burden from
the AeroCom models is 6.3 Tg. Another study of different emission schemes by Spada et al.
(2013) found sea salt burdens in the range from 5.0 to 7.2 Tg. In the light of these studies as
well as the applied correction by the assimilation in CAMSiRA, the simulated sea salt burden
of CR as well as the assimilated burden of MACCRA appears to be too high. The simulated sea
salt emissions of C-IFS were at the upper end of, but still within, the reported range in the
literature (see supplement). This suggests that the high sea salt burden of CR can not entirely
be explained by exaggerated emissions. The underestimation of the loss with respect to other
models must have a played an important role too. On the other hand, the high sea salt burden
of MACCRA was probably caused by an exaggeration of the sea salt emission with an earlier
version of the emissions module.
The desert dust burden in CR was 27 Tg, which was higher than the AeroCom median of 20
Tg. It was strongly reduced by the assimilation in CAMSiRA to 18 Tg. MACCRA had an even
lower desert dust burden of 12 Tg because of the underestimation of the desert dust emissions
scheme used in MACCRA. As in the case of the sea salt, the underestimation of the desert dust
loss by deposition and sedimentation may play an important role in the overestimation of dust
burden in CR.
The strongest relative change in the global burden by the assimilation occurred for sulphate,
which was 1.2 Tg in CR but was 4.7 Tg in CAMSiRA and 3.3 Tg in MACCRA. The respective
AeroCom median value is 2 Tg. Because of the larger extinction per unit mass of sulphate, this
increase in sulphate had a large impact on total AOD, which will be discussed further below.





The organic matter and black carbon burden of CR (0.2 Tg and 2.0 Tg) was increased by the
assimilation to 0.36 Tg and 2.4 Tg respectively. The values agreed reasonably well with the
AeroCom median of 0.21 Tg and 1.76 Tg.
In contrast to the global burden, CR had the lowest global AOD average of 0.13. CAMSiRA
and MACCRA had values of 0.16 and 0.18. The values for CR was close to the median of the
AeroCom models (0.12) but the two reanalyses had a higher value than the highest global
average AOD value of the AeroCom models of 0.15.
The largest fraction of the CAMSiRA AOD came from sulphate, which was strongly increased
by the assimilation. The contribution of sulphate AOD to total AOD was 13% in CR and 43%
in CAMSiRA. Sulphate was also the largest AOD contribution in MACCRA. The global
average of sulphate AOD of CR (0.018) was about half of the AeroCom median (0.034), which
could suggest an underestimation in the global sulphate burden and AOD in CR. On the other
hand, global sulphate AOD of CAMSiRA was 0.06, which was higher than the highest value
of the AeroCom model ensemble (0.051).
As already discussed for the respective burdens, global desert dust AOD and sea salt AOD were
strongly reduced in CAMSiRA compared to CR. In CR sea salt and desert dust AOD
contributed each about 30% to the total AOD, whereas in CAMSiRA the contribution was
reduced to 15% and 19%. The reduction of sea salt by the assimilation was reasonable as the
sea salt burden was above the reported range by Textor (2006) and Spada et al. (2012).
However, the reduction in sea salt was compensated by the increase in sulphate, which became
the most important contribution to total AOD over many parts of the oceans.
The global sea salt burden of MACCRA was higher than in CAMSiRA but similar to CR.
However, a different distribution of the mass within the size classes meant that the resulting sea
salt AOD of MACCRA was 20% higher than CR. MACCRA had the lowest desert dust burden
but differences in the size distribution towards smaller particles meant that the resulting AOD
was slightly higher than CR and 20% higher than CAMSiRA. Black carbon and organic matter
AOD and burden were similar among CAMSiRA, CR and MACCRA.






## 4.2 Spatial patterns of AOD

Figure 8 shows the annual mean of total AOD and AOD for desert dust, sea salt, sulphate, black carbon and organic matter for period 2003–2015 from CAMSiRA and the differences with CR and MACCRA (2003–2012). The global maxima of the total AOD (>0.5) in CAMSiRA were found over areas of desert dust emissions such as the Sahara, the Arabian Peninsula and the deserts of Central Asia. High emissions of black carbon and organic matter from biomass burning sources in tropical Africa and anthropogenic sources in Eastern China and Northern India also produced to AOD maxima on the global scale.

The increase of the global average AOD in CAMSiRA with respect to CR by the assimilation (see section 4.1) occurred in most parts of the globe, in particular over the areas of industrial activity in North America, Europe and East Asia (20–30%) as well as in the polar regions (> 50%), where AOD is generally low. The differences between CR and CAMSiRA, although varying in magnitude, exhibit similar spatial patterns in all seasons, with the largest differences occurring throughout NH in MAM. As discussed in section 4.1 the increase is mostly caused by a wide-spread increase in sulphate AOD. Sulphate AOD was increased in relative terms more strongly over the oceans and higher latitudes. In areas of higher modelled sulphate AOD such as the North America, Europe and Northern Asia and the Arctic the contribution to total AOD changed from 40% to 90%, which made sulphate the by far the most abundant aerosol species in these areas as well as over the Antarctic, which seems unrealistic.

The identified reduction of global desert dust in CAMSiRA with respect to CR was mainly confined to the main desert dust region, where AOD was reduced by to 0.2. As total AOD was dominated by desert dust, total AOD was strongly reduced in these regions, whereas total AOD of CAMSiRA was always higher than CR in the other parts of the globe. The largest relative reduction of desert dust AOD occurred in the remote outflow regions from Australia, Tropical Africa and Eurasia. The reduction of desert dust occurred throughout all seasons with the largest reduction in JJA.

The strongest reduction in sea salt occurred in CAMSiRA with respect to CR occurred over the oceans proportional to the sea salt AOD. Because of the increase in sulphate, the sea salt reduction led only to a small reduction of total AOD over the area of the highest sea salt emissions in the North Atlantic in DJF and over the Southern Ocean in JJA and MAM. The contribution of sea salt AOD to total AOD over most of the ocean was changed from more than





80% in CR to 50% in CAMSiRA in mid- and high latitudes of SH and to 30% over the rest of
the maritime area by the assimilation.
Black carbon and organic matter AOD were reduced in CAMSiRA over tropical Africa where
biomass burning is the largest source on the global scale and also the CO biomass burning
emissions were too high. The black carbon and organic matter AOD values were higher in
CAMSiRA away from the sources where values are generally low. The differences of black
carbon and organic matter AOD between CAMSiRA and CR showed a strong reduction directly
over the areas of intense fire emission in tropical Africa and boreal forest of NH and an increase
in the adjacent outflow regions. This could indicate that the GFAS emissions, as in the case of
CO (see section 3.1), were too high but the atmospheric residence times of the aerosol species
were too short.
Compared to CAMSiRA, MACCRA AOD values were up to 50% (-0.2– -0.3) lower in the
desert dust dominated areas over the Sahara and Central Asia. The largest differences over
North-Africa occurred in JJA and MAM and are an indication that MODIS AOD retrievals are
not available over this regions because of their bright surface (Hsu et al., 2013). The higher
AOD values of CAMSiRA than MACCRA in the desert dust regions might be an improvement
as Cuevas et al. (2015) reported a general underestimation with respect to AERONET
observations in the dust dominated regions of MACCRA.
On the other hand, sea salt AOD over all oceans was much higher in MACCRA than CAMSiRA
and it even exceeded the high sea salt AOD of CR. Despite the higher sea salt AOD, the total
AOD of MACCRA over the oceans was lower than in CAMSiRA because of the overall smaller
sulphate AOD in maritime regions.
In the regions of boreal fire emissions MACCRA was lower during the JJA fire season as well
as in the South American fire season in SON. For the rest of the globe the CAMSiRA, was
about 0.05 lower than the MACCRA, which meant a large relative reduction (>50%) in
particular over the oceans.
The differences between MACCRA and CAMSiRA can mainly be explained with the changes
in the underlying modelling approach and the emissions since the same MODIS AOD retrievals
were assimilated in both reanalyses. Differences in the back ground error statistics may have
contributed to the differences, particularly in the high latitudes.





Figure 9 shows a zonally averaged cross section of the total aerosol mixing ratio of CAMSiRA
and its relative differences of CR and MACCRA. The highest zonal average occurred over the
southern ocean because of the continuous sea salt production, and over the latitudes of the
regions with large desert dust and anthropogenic emissions. Despite the mostly higher AOD
values, CAMSiRA had lower mass mixing ratios than CR throughout the troposphere with the
largest relative differences occurring over the SH mid-latitudes and in the region of intense
convection in the tropics. This is related to a change in the speciation, which was discussed in
section 4.1. CAMSiRA had up to 90% higher values in the stratosphere and Antarctica. The
higher aerosol mixing ratios of CAMSiRA in the upper troposphere were dominated by sulphate
aerosol. MACCRA mixing ratios were considerably higher in relative terms than CAMSiRA
throughout the troposphere with the exception of NH extra-tropical mid- troposphere, caused
by the lower dust emissions in MACCRA, and the SH and tropical stratosphere related to high
sulphate concentrations in CAMSiRA.

## 4.3 Inter-annual variability of AOD

Figure 10 shows time series of average AOD from CAMSiRA, CR and MACCRA for different
regions. To better distinguish the impact of sea salt, the regional AOD is averaged over land
points only. The global average AOD time series are shown separately for land and sea points.
CR and CAMSiRA did not have any significant trends in AOD over the whole globe or any of
the considered regions. There was a good agreement between CAMSiRA and CR in their inter-
annual variability with respect to specific years with higher maxima over South and North
America as well as over Maritime South East Asia and North-Africa. This demonstrates that
despite biases the model was able to reproduce the variability related to fire emissions and wind
driven desert dust suspension. A large relative difference between CR and CAMSiRA occurred
in the Arctic. The CAMSiRA and MACCRA AOD values were almost twice as high as CR and
had a much more pronounced seasonality.
In contrast to the lack of significant trends in CR and CAMSiRA, MACCRA had significant
positive trend over all sea points leading to an increase over 10 years, which was as large as the
seasonal variation over all sea points. Averaged over all land points, the seasonal variation is
much larger than over sea. The agreement in AOD in the monthly means time series was
generally high but MACCRA also showed a significant increasing trend, which was not present
in the other two data sets. Most of this trend in MACCRA was caused by dust AOD, which



increased by 3.7%/yr, and by sea salt AOD, which increased by 1.7%/yr over sea points. We
consider this trend in MACCRA as spurious. It is probably caused by an accumulation of
aerosol mass, which could not be corrected by the assimilation. A reason for the mass
accumulation could be the fact that the MACCRA model did not apply a global mass fixer.
Even if CR and CAMSiRA did not show significant trends in total AOD, sulphate AOD of
CAMSiRA increased significantly by 0.55%/yr and both CR and CAMSiRA had a positive
trend in sea salt AOD of 0.3%/yr. This suggests an artificial accumulation of sulphate by the
assimilation considering that $SO_2$ emissions for aerosol sulphate precursor were constant.
**4.4   Evaluation with AERONET AOD observations**
The AOD at 550 nm was evaluated with observations of the AErosol RObotic NETwork
(AERONET) network. The AERONET is a network of about 400 stations measuring spectral
AOD aerosol with ground based sun-photometers (Holben et al., 1998). The stations are mostly
located over land with a high number of stations situated in North America and Europe. The
global number of stations contributing observations for the evaluation increased from about 60
in 2003 to about 250 in 2014 before it reduced strongly to only a couple of stations at the end
of 2015.
Figure 11 shows time series of the monthly biases of CAMSiRA, MACCRA and CR for the
globe and different regions. Over North America, an area with a high density of AERONET
stations, CR underestimated AOD in general by 0.05 on average. On the other hand, the two
analyses overestimated AOD by about 0.02 but CAMSiRA has marginally smaller biases than
MACCRA. In South America a similar pattern was found only that the average underestimation
of CR and overestimation of CAMSiRA and MACCRA was -0.05 and 0.05 respectively. The
overestimation of CAMSiRA and MACCRA and the underestimation of CR over America
leads to the conclusion that the assimilated MODIS retrievals were biased high against the
AERONET observations in this region as also pointed out in Levy et al. (2010). The underlying
model does not seem to be the cause of the overestimation in CAMSiRA.
Over Europe CAMSiRA had the smallest biases and MACCRA overestimated slightly whereas
CR underestimated the observations. The bias of CR was -0.07 at the beginning of the period
and almost zero at the end. More research is needed to understand this trend in the bias, which
is also apparent in CAMSiRA and MACCRA, but it might be caused by the reduced number of
available stations.



MACCRA had the lowest biases over South East Asia because of small biases in Northern India
and Indochina. It was higher, as almost everywhere, than CAMSiRA and CR. CAMSiRA
underestimated the observations in this region by about 0.05. The underestimation by CR was
bigger and showed a pronounced seasonal cycle. The largest negative biases occurred at the
time of the seasonal minimum in DJF.
The performance for desert dust and sea salt was more difficult to evaluate with AERONET
stations in a robust way because only few stations are available in these regions. The average
bias over Africa showed a strong reduction of the CR peak values, which occurred because of
desert dust outbreaks, by the assimilation. A good example of the successful reduction of dust
by the assimilations was Lake Argyle (16.11.S, 128.75E) in Australia (Figure 11, left).
The AOD AERONET observations over the oceans show generally an overestimation of all
runs, in particular for MACCRA. The bias of the MODIS retrievals with respect to AERONET
(Shi et al., 2011) may be a reason for this overestimation. The comparison with AOD
observation at Mauna Loa Station (19.54 N, 155.58 W, not shown) in the Eastern Pacific
suggests that the low AOD values of CR reproduced the observations best, although still
overestimating them. At Nauru Station (0.52 ° S, 166.9 ° E, Figure 11, right) in the Western
Pacific CAMSiRA match the observations well whereas CR underestimated and MACCRA
overestimated them.
**5   Stratospheric ozone**
The experience from the assimilation of TC and stratospheric profiles retrievals (Inness et al.,
2013, van der A et al., 2015 and Levefer et al., 2015) shows that these observations are sufficient
to constrain stratospheric ozone in the reanalysis. Because almost the same ozone retrievals
were assimilated in CAMSiRA as in MACCRA (see Table 2) most of the differences in the
ozone analyses can be attributed to differences in the ozone simulation of the assimilating
model. For CAMSiRA the Cariolle parameterization (Cariolle and Teyssèdre, 2007) of
stratospheric ozone chemistry and the chemical mechanism CB05 for the troposphere were
used. The tropospheric and stratospheric chemical scheme of the MOZART CTM (Kinnison et
al., 2007) was used for MACCRA.
**5.1   Spatial patterns of TC ozone**
Figure 13 shows the seasonal average TC ozone from CAMSiRA and the difference between
this data set and CR and MACCRA. The differences between CAMSiRA and CR had a




meridional pattern. The assimilation in CAMSiRA increased the total ozone columns in the
tropics and subtropics by up to 25 DU (8%) and it decreased them by 50–70 DU in the NH mid
and high latitudes. The largest reduction occurred in DJF and MAM. Also over Antarctica the
assimilation led to lower values in austral winter (JJA), when TC ozone was reduced by up to
30 DU.
CAMSiRA was about 3–5 DU (1%) lower than MACCRA throughout the globe. Larger
differences of up to 10 DU (2%) were located mainly over tropical land areas. Their shape
suggest that they were partially caused by differences in tropospheric ozone (see section 6.1).
On the seasonal scale, CAMSiRA was about 10 DU lower over Antarctica and the Arctic in the
respective spring seasons MAM and SON.
Figure 14 shows the average ozone partial pressure cross section of CAMSiRA and the relative
differences with CR and MACCRA. The tropospheric part of the figure will be discussed in
section 6.1. The overestimation of CR in the high latitudes of NH and SH was located
predominately in the mid and upper stratosphere at around 20 hPa. The underestimation in the
tropics had the largest values at around 50 hPa.
In the lower and middle stratosphere, i.e. from 70 to 20 hPa, CAMSiRA and MACCRA differed
by less than 5%. Larger differences occurred above 10 hPa where MACCRA was up to 30%
higher than CAMSiRA.
**5.2 Inter-annual variability of TC ozone**
Figure 15 shows area-weighted averages of the monthly TCs for the whole globe, the tropics,
SH and NH mid-latitudes, Antarctica and the Arctic.
In the tropics, CAMSiRA had a significant trend of +0.15%/yr. Although the period of 13 years
is too short to estimate total ozone trends with respect to ozone recovery it is worth noticing
that the number is in good agreement with the estimate of the ozone trend for the period 1995–
2013 by Coldewey-Egbers et al. (2014, see their figure 1), which varies in the tropics between
0.5 to 1.5%/decade. No trends could be found in CR, probably because the climatological
approach applied in the Cariolle scheme is not able to simulate long-term trends. The tropical
trend in MACCRA was 0.25%/yr, which seems too high and there was also a significant trend
in the SH mid-latitudes of 0.65%/yr.
The seasonal range, i.e. the difference between annual maximum and minimum, of TC ozone
in CAMSiRA increased from 10 DU in the tropics to up 150 DU in the Arctic and 100 DU in
Antarctica. As already mentioned in section 5.1, CR was 20% higher than CAMSiRA in NH
mid-latitudes and Antarctica. However, the inter-annual variability agreed reasonably well
between CAMSiRA and CR in SH and MH high and mid-latitudes. For example, the reduced
Arctic ozone spring in 2011 (Manney et al., 2011) and the year-to-year differences in mid-
latitudes found in CAMSiRA were well reproduced by CR.
The ozone hole in Austral spring is the most important feature of seasonal variability over
Antarctica. Remarkably CR, which uses the Cariolle scheme, reproduced the ozone loss during
the ozone hole periods with respect to minimum value and inter-annual variability of TC ozone
very well without assimilating any observations. 2015, 2003 and 2006 were the years with the
deepest ozone holes and 2011, 2013 and 2004 with the shallowest ozone hole both in
CAMSiRA and CR. On the other hand, CR overestimated the average TC ozone during winter
by about 30 DU.
There was generally good agreement between CAMSiRA and MACCRA over all parts of the
globe but MACCRA was on average about 5–10 DU (2%) higher than CAMSiRA. The strong
positive trend of MACCRA in the tropics together with a significant positive trend in the SH
mid-latitudes led to increasing differences of the global average at the end of the MACC period.
Larger difference between MACCRA and CAMSiRA occurred in winter (JJA) over Antarctica,
when MACCRA was up to 25 DU lower than CAMSIRA. The depth of the ozone hole was
slightly deeper in CAMSiRA than in MACCRA.
## 5.3   Evaluation with total ozone retrievals from Dobson sun-photometers
Ozone TCs are observed from the ground with Dobson, Brewer, Point Filter and FTIR
spectrometers. The Dobson instruments provide the longest and best spatial coverage and we
use this data set to evaluate the TC of CAMSiRA, MACCRA and CR. The Dobson instruments
of the WOUDC network are well calibrated and their precision is 1% (Basher, 1982). Factors
that influence the accuracy of the Dobson spectrometer are the temperature dependency of the
ozone absorption coefficient and the presence of $SO_2$.
Figure 16 shows time series of the monthly bias against the Dobson photometer observations
for different regions. Observations of about 50–60 stations were available until 2013 but the
number of stations dropped steadily to about 10 Stations at the end of 2015. CAMSiRA



overestimated the observations in the tropics and the mid-latitudes of both hemisphere on
average by 2 DU whereas the mean bias of MACCRA was about 5 DU larger. In Antarctica
and the Arctic the biases showed a more pronounced seasonal cycle mostly between -10 and 20
DU.
The biases of MACCRA increased in the tropics and the SH-mid latitudes from 2003 to 2008
whereas CAMSiRA and CR did not show an obvious change in the biases until 2012. The
variability of the bias of CAMSiRA amplified at the start of 2013 in NH. As this change in the
bias is not seen at individual stations reporting until the end of 2015, we conclude that the
change is caused by the reduction in stations available after 2013. However, the change of the
assimilated MLS data set (from V2 to V3.4) at the beginning of 2013 (see Table 2).
The biases of CR were much larger than the ones of CAMSiRA, and they had a strong seasonal
cycle. In the tropics CR underestimated the TC by 10 DU in DJF and 0 DU in MAM. The NH
biases were positive and varied between 20–50 DU and in the Arctic between 20–70 DU. Over
Antarctica CR overestimated the observation by 40–60 DU in JJA but the bias was close to zero
or even slightly negative during the time of the ozone hole.

### 5.4 Evaluation with ozone sondes in the stratosphere

The global network of ozone sondes is the most comprehensive independent data set for the
evaluation of the 3D ozone fields from the surface to about 10 hPa, which is the level with the
highest stratospheric ozone volume mixing ratios. The observation error of the sondes is about
±5% in the range between 200 and 10 hPa and -7–17% below 200 hPa (Beekmann et al., 1994,
Komhyr et al., 1995 and Steinbrecht et al., 1996). The number of soundings varied for the
different stations used here. Typically, the sondes are launched once a week but in certain
periods such as during ozone hole conditions launches are more frequent. Sonde launches are
carried out mostly between 9 and 12 hours local time. The global distribution of the launch sites
is even enough to allow meaningful averages over larger areas such North America, Europe,
the tropics, the Arctic and Antarctica.
Figure 17 shows the profiles of the relative biases of CAMSiRA, MACCRA and CR over the
tropics, Antarctica, the Arctic and the NH and SH mid-latitudes for the period 2003–2012. All
available observations were included in the average.
In the tropics, CAMSiRA had a relative bias of mostly below 10% in most of the stratosphere.
MACCRA underestimated the ozone sondes strongly (up to 30%) in the lower stratosphere but



the relative bias of MACCRA was similar or slightly smaller than the bias of CAMSiRA in
most parts of the stratosphere, i.e. in the pressure range from 70 to 20 hPa. CR underestimated
the ozone sondes by up to 20% in the stratosphere up to 30 hPa. The largest underestimation of
CR occurred in the lower and mid stratosphere, where the maximum in ozone partial pressure
is located. In the upper stratosphere above 20 hPa, where the maximum of ozone volume mixing
ratio is located, the relative biases of all data sets were smaller than in the levels below. CR had
almost no bias whereas MACCRA overestimated by up 10%.
Over the Arctic and NH mid-latitudes CAMSiRA and MACCRA agreed well with the sondes
in the whole stratosphere with relative biases below 5%. The absolute biases of CAMSiRA
were slightly smaller than the biases of MACCR in particular in the lower stratosphere and
upper troposphere. CR overestimated the ozone observations by up to 25% in the stratosphere
and upper troposphere over the Artic and up to 20% in the NH mid-latitudes. The relatives
biases of CR tended to be slightly smaller in the mid stratosphere (50 hPa) than in the upper
and lower stratosphere.
Over SH-mid latitides and Antarctica the annual biases in the stratosphere were slightly smaller
in CAMSIRA than MACCRA but for both reanalyses they were below 10%. As over the Arctic,
the absolute tropospheric biases, with the exception of the surface values, were smaller in
MACCRA since CAMSiRA showed an underestimation of about 10%. CR had a stronger
underestimation in the lower and upper stratosphere.
As the process of the ozone-hole formation cannot easily be demonstrated with annual means,
Figure 18 shows the monthly mean profile from August to November over Neumayer Station
(70.7° S, 8.3° W). The two reanalysis agreed very well with the observations: vertical level and
magnitude of the ozone profile at the end of the austral winter in August, the ozone depletion
in September and October and the closure of the ozone hole starting in the upper stratosphere
were well captured because of the assimilation of TC and limb-sounders profiles.
In contrast, CR showed a strong overestimation in August in the middle and lower stratosphere.
Ozone in the upper stratosphere in September was underestimated in CR because of an
exaggerated depletion whereas ozone was overestimated in the lower stratosphere. In the
following months CR ozone remained too high in the lower stratosphere and too low in the
upper troposphere but the resulting TCs matched the observations in a reasonable way (see
Figure 16)



## 5.5   Evaluation with the GOZCARDS ozone product in the upper stratosphere

Ozone sondes do not provide accurate measurements above 10 hPa. The ozone bias profiles shown in Figure 17 indicate higher values of MACCRA in the upper stratosphere and mesosphere, i.e. from above 10 hPa to the model top of 0.1 hPa. Although the ozone mass in this region is relatively small, the high values of the mixing ratios have a large impact on the radiative transfer and the associated heating rates. To investigate the biases in that region we used the Global OZone Chemistry And Related trace gas Data records for the Stratosphere (GOZCARDS) product (Froidevaux et al., 2015). It consists of merged SAGE I, SAGE II, HALOE, UARS and Aura MLS, and ACE-FTS data from late 1979 to 2012. SAGE II is used as the primary reference in the merging procedure for the instruments. For most of the CAMSiRA period, i.e. from 2004 onwards, Aura MLS and ACE-FTS are the dominating instruments in the upper stratosphere. Tegtmeier et al. (2013) showed that ozone retrievals from various instruments show a considerable spread in the upper stratosphere. ACE-FTS is biased high above 10 hPa and biased low below 10 hPa against the median of various retrievals.

Figure 19 shows cross sections of the GOZCARDS product and relative bias of CAMSiRA, MACCRA and CR in the vertical range from 50–0.3 hPa. In the region from 10–5 hPa MACCRA had a positive bias of 10–15% in the tropics and mid-latitudes, which has already been reported in Inness et al. (2013). About half of the 10 DU higher TCs in MACCRA compared to CAMSiRA were caused by this overestimation in the levels above 10 hPa. The biases of CAMSiRA in that region were smaller and vary between 2.5 and -2.5%. CAMSiRA underestimated the GOZCARDS data between 5 and 1 hPa by up to 7%, whereas MACCRA slightly overestimated. In the lower mesosphere MACCRA underestimated the ozone concentrations by up to 30%.

CR had very similar biases as CAMSiRA above 5 hPa in the tropics and mid-latitudes. This means that the assimilation of observations had already little influence in this region even if no increments were added during the CAMSiRA assimilation above 1 hPa. Below 10 hPa the cross section of the bias shows the already discussed strong overestimation of CR in the mid and higher latitudes, which was largest in relative terms at around 20–15 hPa and the underestimation in the tropics, which was largest at around 50 hPa.



## 6 Tropospheric ozone


Correcting tropospheric ozone by the assimilation of TC and stratospheric ozone profiles
remains a challenge because the observations are dominated by the high stratospheric mixing
ratios (Wagner et al., 2015). The modelled ozone fields as well as the specification of the
vertical background error correlation have therefore a large impact on the analysed tropospheric
ozone fields (Inness et al., 2015).

### 6.1 Spatial patterns of ozone at 850 hPa


We focus the discussion of the seasonal spatial patterns of monthly mean tropospheric ozone
mole fraction to the 850 hPa pressure level values but we also discuss tropospheric ozone at
500 and 200 hPa in the section 6.2 and comparisons with ozone sondes for different
tropospheric layers in section 6.3. Figure 20 shows the seasonal means of CAMSiRA and the
differences with CR and MACCRA at 850 hPa. Extratropical NH ozone values of CAMSiRA
were mostly in the range from 35–55 ppb. The season of the maximum was MAM, when values
were about 20 ppb higher than in the seasonal minimum in DJF. Regional maxima of over 60
ppb were situated over the East Asia and the Arabian Peninsula. JJA was the season when the
highest values occurred over the areas of the regional maxima. In this season an additional
regional maxima occurred over tropical Africa. The SH values were generally below 35 ppb.
The seasonal maximum was in Austral spring (SON) and the minimum in Austral summer and
late autumn (SON).
CR was about 2–4 ppb higher than CAMSiRA in most parts of the globe. Only in the higher
latitudes of SH as well as over the biomass burning regions in Africa, South America and
Maritime South East Asia, CAMSiRA was up to 4 ppb lower than CR. The biggest large-scale
reduction by the assimilation in NH occurred in DJF and the biggest increase in SH in SON.
The largest absolute increases of CAMSiRA of up to 10 ppb occurred over the Southern end of
the Arabian Peninsula at the time of the seasonal maximum in JJA. This was the only local
maximum in CAMSiRA that was increased by the assimilation.
Tropospheric ozone was the only considered species for which the differences between
CAMSiRA and MACCRA were larger than the difference between CAMSiRA and CR. This
indicates the importance of the chemistry model parameterization and the limitations of the data
assimilation in this respect. In the extra-tropics of NH and SH, CAMSiRA was 2–5 ppb lower
than MACCRA with an increasing difference towards the poles. The largest difference occurred





in NH summer in JJA. CAMSiRA was up to 10 ppb lower than MACCRA over the continents
in the tropics. On the other hand, CAMSiRA had higher values than MACCRA over the tropical
oceans, the Sahara as well as at the location of the strong maximum over the Arabian Peninsula,
which was not present in MACCRA. The strong land-sea contrast in the differences could be
caused by (i) a different efficiency of deposition over the oceans, (ii) the discussed differences
in biomass burning emissions and (iii) differences in the chemistry treatment (e.g. the isoprene
degradation scheme).
The vertical distribution (see Figure 14) of the mean ozone partial pressure in the troposphere
shows that CAMSiRA was lower than CR in the whole troposphere apart from the tropical
upper troposphere, where it was up to 10% higher, as well as below 500 hPa in the SH
troposphere. Compared to MACCRA, CAMSiRA was up to 20% higher in the middle and
upper troposphere in the tropics and subtropics but increasingly lower towards the surface.
## 6.2  Inter-annual variability
Estimating and understanding tropospheric ozone trends have been studied widely in the
literature, as reviewed in Cooper et al. (2014) and Monks et al. (2015). Factors that influence
the inter-annual variability and trends of tropospheric ozone are changes in anthropogenic and
biomass burning emissions, the stratosphere-troposphere exchange and the variability of the
meteorological fields. The observed trends vary strongly because these different factors are not
uniform in space and time. Trends are often confined to specific seasons or levels. Positive
trends are more common than negative trends and are found over Europe and North America
during spring (Cooper et al., 2014).
Figure 21 shows time series of average ozone volume mixing rations over selected regions and
pressure levels at 850, 500 and 200 hPa. It is beyond the scope of the paper to investigate the
robustness of the trends in CAMSiRA in detail. But it is worth noting that there were only
positives trends in the considered region at 850, 500 and 200 hPa in CAMSiRA. The trends
varied between 0-1.1%/yr, with a global mean of 0.5%/yr. Many of these trends were
significant. CR also had mostly positive but much smaller trends with a global mean of
0.17%/yr. The only significant trend in CR of 0.35%/yr was found over East-Asia and the
corresponding trend in CAMSiRA had the same value. Focusing over Easter China, Verstraeten
et al. (2015) find a trend of about 1.2%/yr between 2005 and 2010, which is considerably larger
than the trend in CAMSiRA and CR.





The time series in Figure 21 show that the higher values in NH of CR with respect to CAMSiRA
occurred in the entire troposphere. In the lower and mid troposphere CAMSiRA was lower than
CR especially during the seasonal minimum. In the tropics, CR and CAMSiRA agreed well at
850 hpa, CR was slightly higher at 500 hPa and about 5 ppb lower than CAMSiRA at 200 hPa.
At this level CAMSiRA had a significant trend of 0.95%/yr in the tropics, which was not present
in CR. More detailed studies are needed to confirm the realness of this upper tropospheric trend
in CAMSiRA.
A more detailed inspection of the time series shows that from the start of 2013 CR and
CAMSiRA agree to higher degree than before in the middle and upper part of the troposphere
in NH. The agreement is most likely caused by a reduced correction by the assimilation in the
NH troposphere in this period. In early 2013 the assimilated MLS ozone retrieval switched from
version V2 to the NRT V3.4 product (see Table 2), which had different levels and observations
errors. The discontinuation of the MIPAS in spring 2012 do not seem to be the reason for this
behaviour.
The year-to-year variability of tropospheric ozone from MACCRA did often not resemble that
of CAMSiRA. In NH at 850 hPa (most prominently seen in the Arctic) MACCRA had
increasing values until 2008 after which they dropped to the values of CAMSiRA. This drift of
MACCRA and the associated negative trends are not realistic (as confirmed in section 6.3).
They were caused by applying the variational bias correction scheme to MLS data in MACCRA
(see Inness at al. 2013 for more details). The agreement between CAMSiRA and MACCRA
increases with increasing height in the extra-tropics but in the tropics MACCRA showed a
much stronger trend at 200 hPa than CAMSiRA.
**6.3**   *Evaluation with ozone sondes in the troposphere*
Figure 22 show time series of seasonal biases in pressure ranges representing the lower, middle
and upper troposphere from 6 different ozone sonde sites. The selected stations had at least one
observations for each month of the 2003-2105 period and are examples for Europe (De Bilt),
North America (Huntsville), the tropics (Nairobi), the Arctic (Ny-Ålesund) and Antarctica
(Neumayer Station). To present South-Asia we chose Hong Kong Observatory, which had
complete cover from 2003-2012. These individual time series depend on the specific
characteristics of the individual stations and are therefore less representative than the averages
over the gridded data sets shown in section 6.2.





In the lower troposphere (950-700 hPa) over DeBilt, Huntsville and Nairobi, CR and
CAMSiRA had seasonal biases in the mostly in the range of -7–7 ppb. In the polar regions at
Neumayer Station and Ny-Ålesund both CR and CAMSiRA underestimated the observations.
At all locations CAMSiR was lower in the lower troposphere than CR, which meant that
CAMASiRA had mostly a larger absolute bias than CR. At Hong Kong Observatory both
CAMSiRA and CR overestimated the observations with biases in the range between 0-10 ppb.
In the middle troposphere the absolute biases of CAMSiRA and CR were of the same magnitude
but of different signs. In the upper troposphere CR overestimated the observations by about 10
ppb whereas the bias of CAMSiRA remained below 5 ppb. The overestimation of CR is
probably caused by the influence of the stratosphere where CR was too high (see section 5.4).
Over Nairobi the biases of CR and CAMSiRA were very similar in all levels but CAMSiRA
had overall lower biases in the lower troposphere. In the pressure range 400–300 hPa in the
tropics the impact of stratospheric biases on CR is less strong because of the higher tropopause
height in this region.
The biases for all three data sets at Ny-Ålesund, Hunstville and Hong Kong Observatory
showed a pronounced seasonality in the middle and upper troposphere. At Huntsville the spring
maximum was especially overestimated, i.e. it occurred 2-3 month too early. At Ny-Ålesund
the overestimation was caused by too high values in summer and autumn. Over Hong Kong
Observatory the pronounced observed spring maximum was not well reproduced.
As already discussed in section 6.2, the characteristics of the bias of CAMSiRA changed at the
start of 2013 mainly in the upper parts of the NH troposphere but also throughout the
troposphere over higher latitudes. In this period the CAMSiRA biases resembles much more
the bias of CR which often mean an increase in the average values, which could cause a spurious
enhancement of positive trends.
At Neumayer Station CAMSiRA increased in a step-wise manner already at the start of 2012,
which changed the bias from an underestimation to a slight overestimation together with an
increased seasonality. This behaviour could be caused by the discontinuation of MIPAS in
spring 2012 (see Table 2). Although the MIPAS retrievals were only stratospheric profiles, the
combined assimilation with total column retrievals can trigger a correction in the troposphere
(Flemming et al., 2011).





MACCRA had a less stable bias than CAMSiRA. In the lower and mid-troposphere biases from
2006–2008 were much higher than in the rest of the period, when they resembled more the
biases of CAMSiRA and CR. This confirms that the discussed inter-annual variability of
MACCRA seem less realistic than that of CR and CAMSiRA.
It should be noted that both MACCRA and CAMSiRA suffered from larger than typical
negative biases in the NH in the first half of 2003, which can probably be explained by biases
in the initial conditions.
### 6.4   *Evaluation with Airbase Ozone surface observations*
The AirBase and EMEP databases host operational air quality observations from different
national European networks. All EMEP stations are located in rural areas, while Airbase
stations are designed to monitor pollution at different scales. Stations of the rural regime can
capture the larger scale signal in particular for $O_3$, which is spatially well correlated (Flemming
et al., 2005). Therefore EMEP stations and only rural Airbase stations were used in the
evaluation to account for the model resolution of C-IFS.
Figure 23 shows the average diurnal cycle for each season of the observed values and
CAMSiRA, CR and MACCRA. CR and CAMSiRA were very similar and matched well the
shape of the observed diurnal cycle. However there was a constant bias of about 5 ppb in MAM
and DJF. CR had slightly smaller biases than CAMSiRA in JJA in the afternoon. MACCRA
had a larger diurnal range because the day-time values were higher than the ones of CAMSiRA.
This meant smaller day-time biases in MAM and DJF and hence a smaller seasonal bias for
MACCRA. But it also led to a considerable (10 ppb) day time overestimation in JJA and a
smaller overestimation in SON as well as a less well fit with the shape of the observed diurnal
cycle in all seasons.
The winter and spring underestimation of CAMSiRA and CR has already been reported in
Flemming et al. (2015). To investigate the possible causes of this seasonal bias Figure 24 shows
the average seasonal cycle at the surface at the EMEP-AirBase stations and in the lower
troposphere (950–750 hPa) over ozone sonde stations. The differences between CAMSiRA, CR
and MACCRA were more pronounced in the lower troposphere than at the surface. This
indicates again that the assimilation has little influence on the surface values. CR matched the
observations in the lower troposphere well in all seasons apart from SON, when it
overestimated. MACCRA had similar biases as CR but overestimated additionally in JJA and



especially over southern Europe, as shown in Katragkou et al. (2015). CAMSiRA
underestimated throughout the year with the exception of SON. As the patterns of the seasonal
biases were different in the lower troposphere and at the surface, we conclude that the winter
and spring-time bias at the surface is not predominately caused by tropospheric biases. It is
more likely that the simulation of surface processes such dry deposition and titration by freshly
emitted NO are the reasons for this bias at the surface.

## 7    Summary and conclusions

CAMSiRA is a new reanalysis data set of aerosol, CO and ozone for the period 2003–2015. It
has been produced by assimilating satellite retrievals of AOD, TC CO as well as TC and
stratospheric ozone profile retrievals from various sensors in C-IFS using the ECMWF 4D-
VAR approach. A similar set of observations was assimilated in MACCRA, a previous
reanalysis data set for the period 2003–2012. A control run with C-IFS (CR) without the
assimilation of AC observations was carried to infer the impact of the assimilated observations.

### 7.1    CAMSiRA compared to MACCRA

Compared to its predecessor MACCRA, CAMSiRA had smaller biases of surface and lower
tropospheric CO as shown by the comparison with MOZAIC/IAGOS CO profiles and NOAA-
GMD CO flask observations. However, MACCRA had lower CO biases in NH mid and upper
troposphere with respect to the MOZAIC/IAGOS CO profiles. The biases of TC ozone against
the WOUDC Dobson sun photometers where reduced from 5–10 DU in MACCRA to 0–5 DU
in CAMSiRA. The biases of CAMSiRA against AERONET AOD observations were lower in
most parts of the globe with the exception of South East Asia. A larger improvement was the
elimination of the positive bias of upper stratospheric ozone in MACCRA as shown by the
comparison with the GOZCARDS ozone product. CAMSiRA also had a better agreement with
the shape of the mean observed diurnal cycle of AIRBASE ground-level ozone observations in
Europe in all seasons but winter and spring time seasonal values were still underestimated by 5
ppb. We attribute all the aforementioned differences between CAMSiRA and MACCRA, which
were mainly improvements, to the change of the assimilating model, which was the coupled
system IFS-MOZART for MACCRA and C-IFS with updated aerosol parameterizations for
CAMSiRA.
Progress achieved by changes to the assimilated observations was a noteworthy improvement
of the temporal consistency of the tropospheric CO and ozone fields in CAMSiRA. The



assimilation of IASI CO in MACCRA from 2008 onwards had led to a decrease in the TC CO
values because of the biases against the MOPITT data set, which was assimilated during the
whole period. Consequently, the MACCRA CO fields in the mid- and high latitudes of both
hemispheres showed strong negative trends which were not in agreement with linear trends
estimated from CO flask surface observations. On the other hand, the linear trends of
CAMSiRA agreed well with the observed trends, which were close to zero in SH and reached
values of about 2 ppb/yr in the NH mid and high latitudes. The mid and upper tropospheric
ozone fields of MACCRA suffered from an increase in the period 2004–2008 caused by a
applying disproportionate application of the inter-instrument bias correction to the MLS column
retrievals, which was corrected for CAMSiRA (Inness et al., 2015).
A discontinuity in the upper and middle tropospheric ozone field was noted for CAMSiRA after
January of 2013 and was due to a change in version of the assimilated MLS ozone retrievals.
Although this change in CAMSiRA did not mean an increase in the bias, it has to be considered
when trends of tropospheric ozone fields are to be calculated from the CAMSiRA data set.
The AOD in CAMSiRA was about 0.01 lower than MACCRA in most parts of the globe,
mainly because of a 50% lower burden of sea salt in CAMSiRA. CAMSiRA had higher AOD
values over the desert dust emitting regions in North-Africa and the global desert dust burden
was higher in CAMSiRA. CAMSiRA had 25% higher AOD contribution by sulphate than
MACCRA, which is currently under scrutiny.
**7.2   CAMSiRA compared to CR**
The comparison with CR showed that the assimilation led to a clear improvement for CO, AOD
and TC ozone as well as stratospheric and upper tropospheric ozone.
The assimilation of MOPITT CO increased the values in the NH mid-latitudes more in the
beginning of the period, which could indicate a stronger underestimation of the anthropogenic
emissions in this period as well as an overestimation of the trend in the emissions. The tropical
and SH values were reduced by the assimilation, which may indicate an overestimation of the
biomass burning emissions in this region. However, the rather zonally homogeneous CO
differences between CR and CAMSiRA suggest that not only biases in the fire emissions but
also of the lifetime and the CO transport need to be investigated further.
The Cariolle scheme for stratospheric ozone, which was used in C-IFS, suffered from a large
overestimation of NH mid-and high latitude stratospheric ozone (50–100DU) and an



underestimation in the tropics (-20 DU). These biases were corrected by the assimilation and
the resulting biases of CAMSiRA were of 5 DU and lower. Also in the SH high-latitudes the
Cariolle scheme overestimated the mean TCs especially in JJA by up to 30 DU but the depth
and the year-to-year variability of the ozone hole was well reproduced by CR. Nevertheless,
CAMSiRA had more realistic TCs and profiles than CR during the annual ozone hole events.
The assimilation had only little impact on the ozone values at the surface and in lower
troposphere, where the biases of CAMSiRA where sometimes even slightly larger than of CR.
The small influence could be explained by the fact, that dry deposition velocities and important
ozone precursors such as $NO_x$ were not constrained during the assimilation process. Also
contributing was the fact that no direct tropospheric ozone observations were assimilated. The
assimilation was more beneficial in the upper troposphere, where the stratospheric influence is
more important.
CAMSiRA had about 0.05 higher AOD values than CR apart from the desert dust emission
regions, where the assimilation strongly reduced the modelled values. CAMSiRA tended to
slightly overestimate the AERONET AOD observations and CR to underestimate but the
overall biases of CAMSiRA were smaller.
Despite moderate differences in AOD, CR and CAMSiRA had considerable differences in the
aerosol speciation. The global annual sea salt burden by C-IFS in CR of 15 Tg was considerably
higher than the result of other modelling studies (Textor et al., 2006 and Spada et al., 2012).
Less efficient loss processes may have played a large role in this overestimation. The
assimilation strongly reduced the sea salt burden in CAMSiRA to about half of the value in CR.
Also the global desert dust burden was reduced by 25% by the assimilation leading to lower
total AOD values over the desert dust emissions regions of Sahara, Australia and Middle Asia.
Despite the fact that CAMSiRA had a 30% smaller global aerosol burden, its average global
AOD was about 10% higher than the one of CAMSiRA. This was caused by a strong increase
in sulphate in CAMSiRA. The optical properties and assumed size distribution of sulphate make
extinction more efficient for the same amount of mass. Sulphate became the dominant
contribution to AOD in the regions away from the main aerosol emissions. The strong
contribution of sulphate may have partly compensated for the inadequate representation of other
secondary aerosols in C-IFS. However its magnitude and spread over the whole globe seems
excessive. It might be caused by the lack of strong loss processes in the free troposphere as well
as biases in the assimilated observations over the open oceans. As the CR underestimates the



assimilated AOD, the aerosol mass is increased during the assimilation, initially by the same
relative amount for all components. However, a longer life-time of sulphate causes a longer
lasting change compared to the other aerosol species, which made sulphate the dominating
aerosol. This distortion of the speciation can not be corrected by the assimilated MODIS AOD
retrievals, which do not contain information about the speciation.

### 7.3   Recommendations for future AC reanalysis

CAMSiRA is considerable improvement over MACCRA especially with respect to the
temporal consistency. To further improve on this important aspect, one should make sure that
consistent input emission data sets and assimilated observations are used. Changes in the
assimilated observations, such as the version change in the MLS after 2012 should be avoided.
The use of MEGAN simulated biogenic emissions for the whole period is advisable even if no
related jumps were detected in this study. To ensure consistency between the aerosols and
chemistry components, the same $SO_2$ emissions should be used.
As improvements to lower tropospheric ozone by assimilating current satellite observations are
difficult to achieve, emphasis needs to be put on the improved simulation of chemistry and dry
deposition. The assimilation of tropospheric ozone column retrievals as well as of tropospheric
$NO_2$ may further help to improve the ground level ozone in the reanalysis. To further develop
the C-IFS assimilation system to allow the correction of ozone-precursor emissions could be an
important next step towards an improved tropospheric ozone analysis.
The high sulphate burden introduced by the assimilation can perhaps be avoided by (i) the
introduction of more intensive loss processes in the free troposphere, (ii) an increase of the
organic matter to better represent non-accounted SOA components and (iii) changes to the
vertical structure of the background errors to avoid the accumulation of aerosol mass away from
the surface. In general, any modelling improvements for a better speciation will reflect in a
more realistic aerosol analysis and a better exploitation of the available observations. If possible
the latest reprocesses MODIS AOD dataset should be used (collection 6).
In CAMSiRA and MACCRA the aerosol and chemistry schemes were independent. A better
coupling between the two and the meteorological simulation is desirable. For example the use
of aerosol to modulate photolysis rates and heterogeneous uptake on aerosol as well as the
simulating the impact on aerosols and ozone within the radiation transfer calculation of IFS will
be important next steps.




**Data access**

The CAMSiRA data are freely available. Please contact copernicus-support@ecmwf.int

**Acknowledgments**

CAMS is funded by the European Union's Copernicus Programme. The GOZCARDS data
were obtained from the NASA Goddard Earth Science Data and Information Services Centre.
We are grateful to the World Ozone and Ultraviolet Radiation Data Centre (WOUDC) for
providing ozone sonde and Dobsen-photometer observations. We thank the Global
Atmospheric Watch programme for the provision of CO and ozone surface observations. We
thank the European Environmental Agency for providing access to European ozone
observations in the AirBase data base. We also thank the MOZAIC (Measurements of OZone,
water vapour, carbon monoxide and nitrogen oxides by in-service AIrbus aircraft) and IAGOS
(In-Service Aircraft for a Global Observing System) programmes for providing CO profile
observations.




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

|  | MACCRA | CAMSiRA |
|---|---|---|
| Period | 01/2003–12/2012 | 01/2003–12/2015 |
| Horizontal resolution | 80 km (T255) | 110 km (T159) |
| Vertical resolution | 60 layers from surface to 0.1 hPa | *as MACCRA* |
| Anthropogenic Emissions | MACCity (trend: ACCMIP + RCP 8.5), AEROCOM | *as MACCRA* & CO emission upgrade (Stein et al., 2014) |
| Chemistry module | MOZART-3 | C-IFS CB05 / Cariolle ozone |
| Assimilated CO observations | MOPITT (V4) & IASI (from 2008 onwards) | MOPITT (V5) & updated error statistics (Inness et al., 2015) |



| | | |
|---|---|---|
| Assimilated ozone observations | SBUV-2, OMI, MLS, GOME-2, SCIAMACHY, GOME, MIPAS (01/2003–06/2004) | *as MACCRA* & MIPAS (2003–2012) |
| Ozone MLS bias correction | On | Off |
| Assimilated AOD observations | MODIS (Aqua and Terra) + VarBC | *as MACCRA* |
| Fire emissions | GFED (2003–2008) and GFAS v0 (2009-2012 ) | GFAS v 1.2 (2003–2015) |
| IFS model version | CY36R2 | CY40R2 |
| Assimilation method and model | ECMWF 4D-VAR | *as MACCRA* |
| Meteorological observations assimilated | ECMWF RD setup (satellites, sondes, surface ) | *as MACCRA* |

*Table 1 Important commonalities and differences between MACCRA and CAMSiRA*







| Instrument | References | Version | Period | Type | Data usage |
|---|---|---|---|---|---|
| MOPITT Terra | Deeter et al. (2011) | V5 TIR NRT | 20030101–20121218 From 20121219 | CO TC | 65N–65S QC=0 |
| GOME ERS-2 | Munro et al. (1998) | | 20030101–20030531 | O3 profile | 80N–80S SOE>15, QC=0 |
| GOME-2 Metop A | Hao et al. (2014) | NRT GDP4.4 NRT GDP4.7 | 20120901–20130714 From 20130715 | O3 TC | SOE>10 QC=0 |
| GOME-2 Metop B | Hao et al. (2014) | NRT GDP4.7 | From 20140101 | O3 TC | SOE>10 QC=0 |
| MIPAS Envisat | von Clarmann et al. (2003, 2009) | NRT CCI | 20030101–20040326 20050127–20120331 | O3 profile | QC=0 |
| MLS Aura | Froidevaux et al. (2008) | V2 NRT V3.4 | 20040808–20121231 From 20130107 | O3 profile | QC=0 |
| OMI Aura | Liu et al. (2010) | V003 NRT | 20041001–20121231 From 20130101 | O3 TC | SOE>10 QC=0 |
| SBUV/2 NOAA-16 | Bhartia et al. (1996) | V8 | 20040101–20081020 | O3 PC 6 layers | SOE>6 QC=0 |
| SBUV/2 NOAA-17 | Bhartia et al. (1996) | V8 | 20030101–20121130 | O3 PC 6 layers | SOE>6 QC=0 |
| SBUV/2 NOAA-18 | Bhartia et al. (1996) | V8 | 20050604–20121217 | O3 PC 6 layers | SOE>6 QC=0 |
| SBUV/2 NOAA-19 | Bhartia et al. (1996) | V8 | From 20090100 | O3 PC 6 layers | SOE>6 QC=0 |
| SCIAMACHY Envisat | Eskes et al. (2012) | CCI | 20030101–20120408 | O3 TC | SOE>6 QC=0 |
| MODIS / Terra | Remer et al. (2005) | Col.5 NRT Col.5 | 20030101–20080731 From 20080801 | AOD 550nm | 70N-70S |
| MODIS / Aqua | Remer et al. (2005) | Col.5 NRT Col.5 | 20030101–20080731 From 20080801 | AOD 550nm | 70N–70S |

*Table 2 Assimilated satellite observations in CAMSiRA*




| Area | Coordinates |
|------|-------------|
| North America | 165°W-55°W, 25°N–75°N |
| Europe | 10°W–45°E, 38°N–70°N |
| East Asia | 90°E–150°E/10°N–55°N |
| South America | 82°W–30°W/40°S–15°N |
| Tropical Africa | 15°W–55°E/10°S–20°N |
| Northern Africa | 15°W–55°E/20°N–35°N |
| Maritime South East Asia | 90°E–150°E/10°S–10°N |
| Tropics | 23°S–23°N |
| Arctic | 60°N–90°N |
| Antarctica | 90°S–60°S |
| NH mid latitudes | 30°N–60°N |
| SH mid-latitudes | 60°S–30°S |

*Table 3 Coordinates of regions*







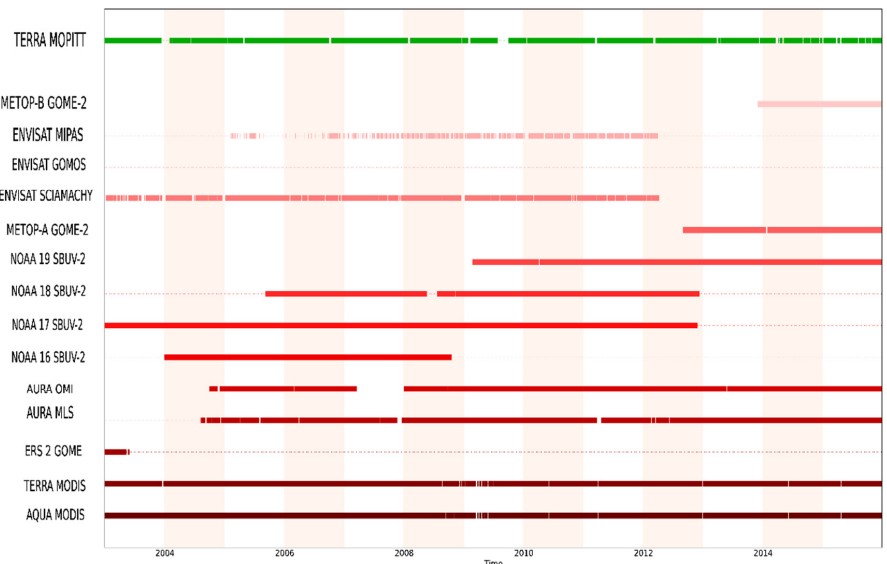


Figure 1 Time line of assimilated AC satellite retrievals from different instruments
assimilated in CAMSiRA (see Table 2)






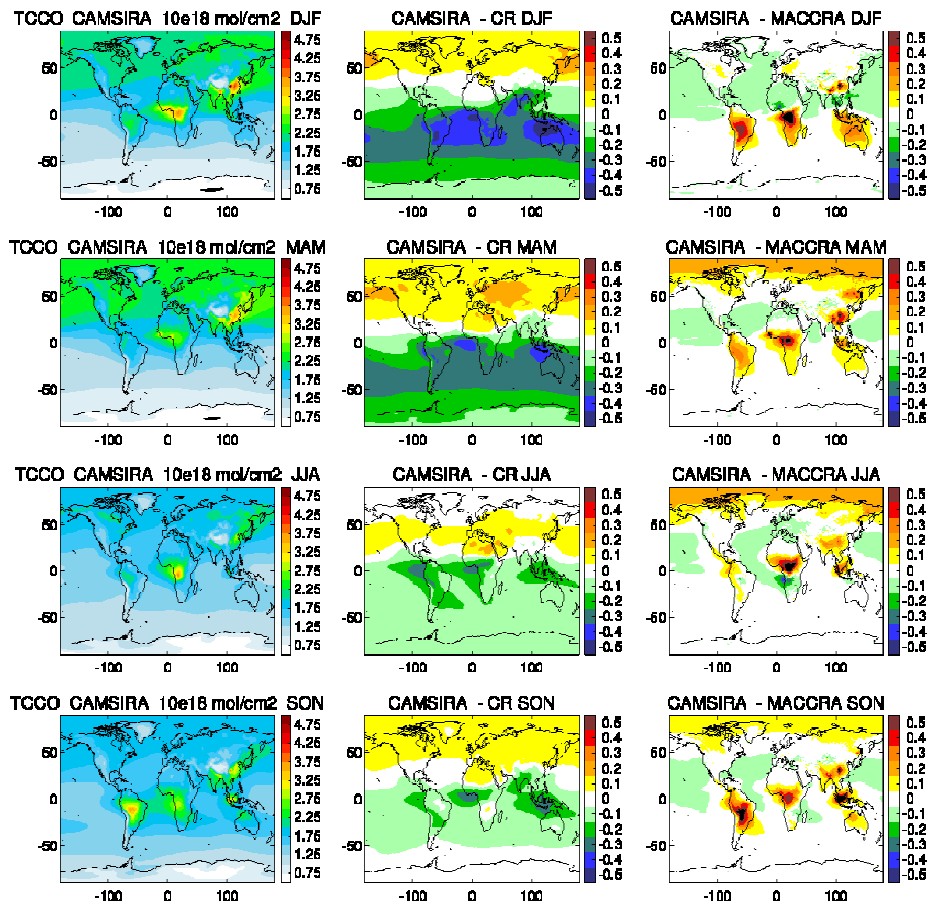


*Figure 2 Average TC CO ($10^{18}$molecules/cm$^2$) of CAMSiRA (2003–2015, left) and difference against CR (2003–2015, middle) and MACCRA (2003–2012, right) for the seasons DJF (row 1), MAM (row 2), JJA (row 3) and SON (row 4).*






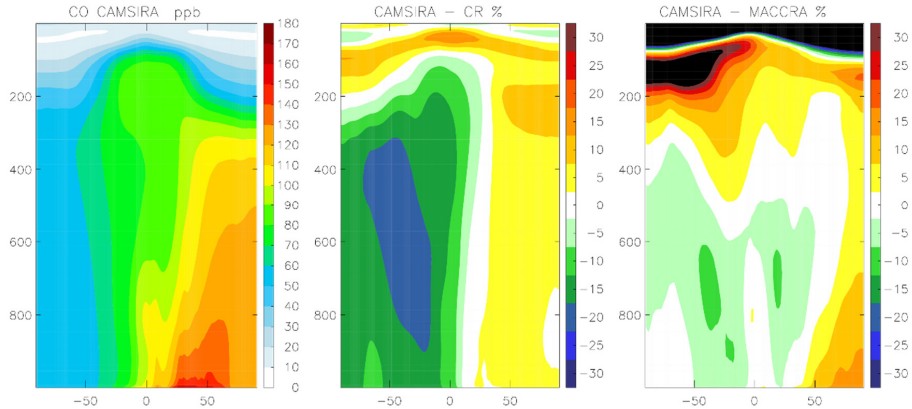


*Figure 3 Zonally averaged CO cross section of CAMSiRA (ppb) (2003–2015, left) and*
*relative difference (%) against CR (2003–2015, middle) and MACCRA (2003–2012, right).*





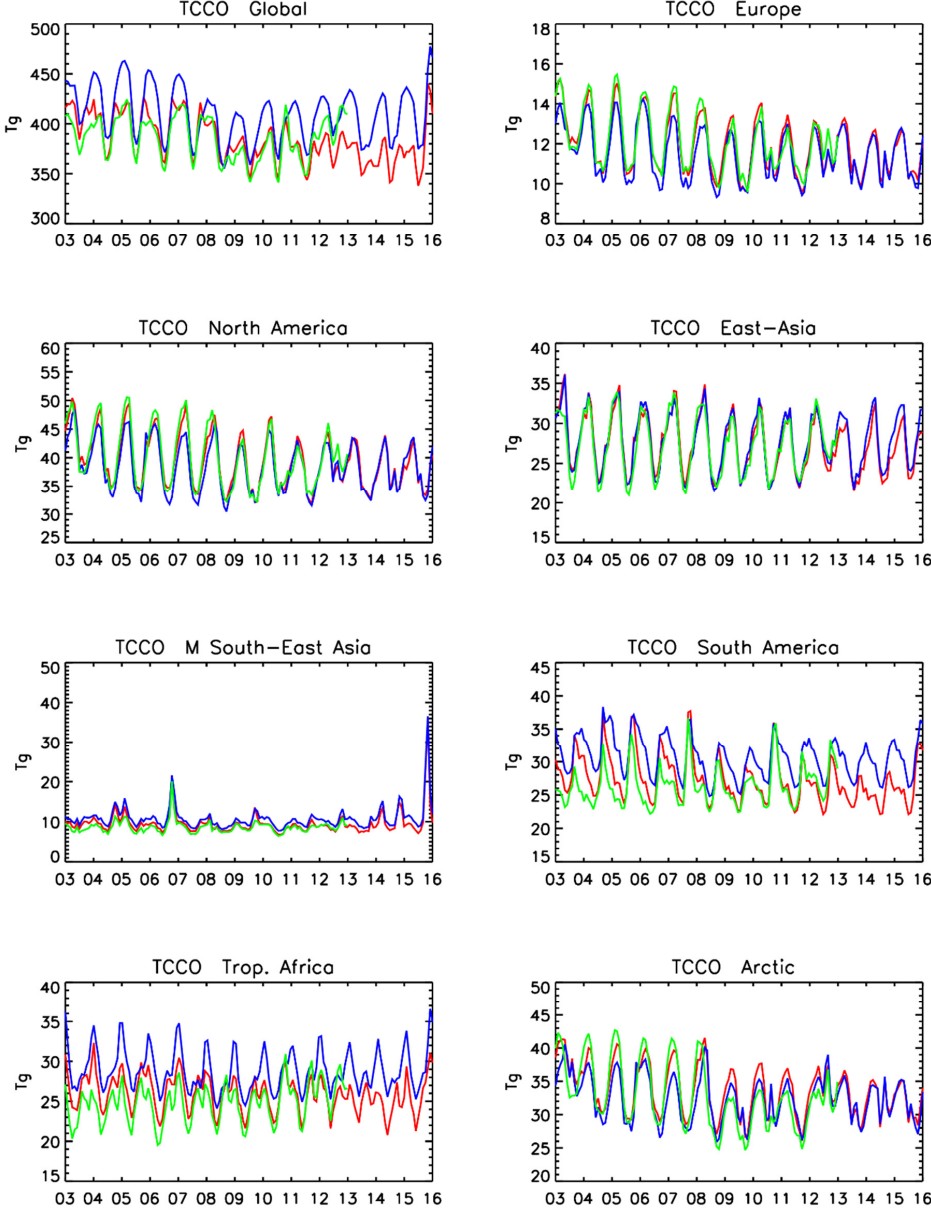


*Figure 4 Time series of monthly mean CO burden (Tg) over different regions (see Table 3)*
*for the period 2003–2015 from CAMSiRA (red), CR (blue) and MACCRA (green, 2003–*
*2012).*





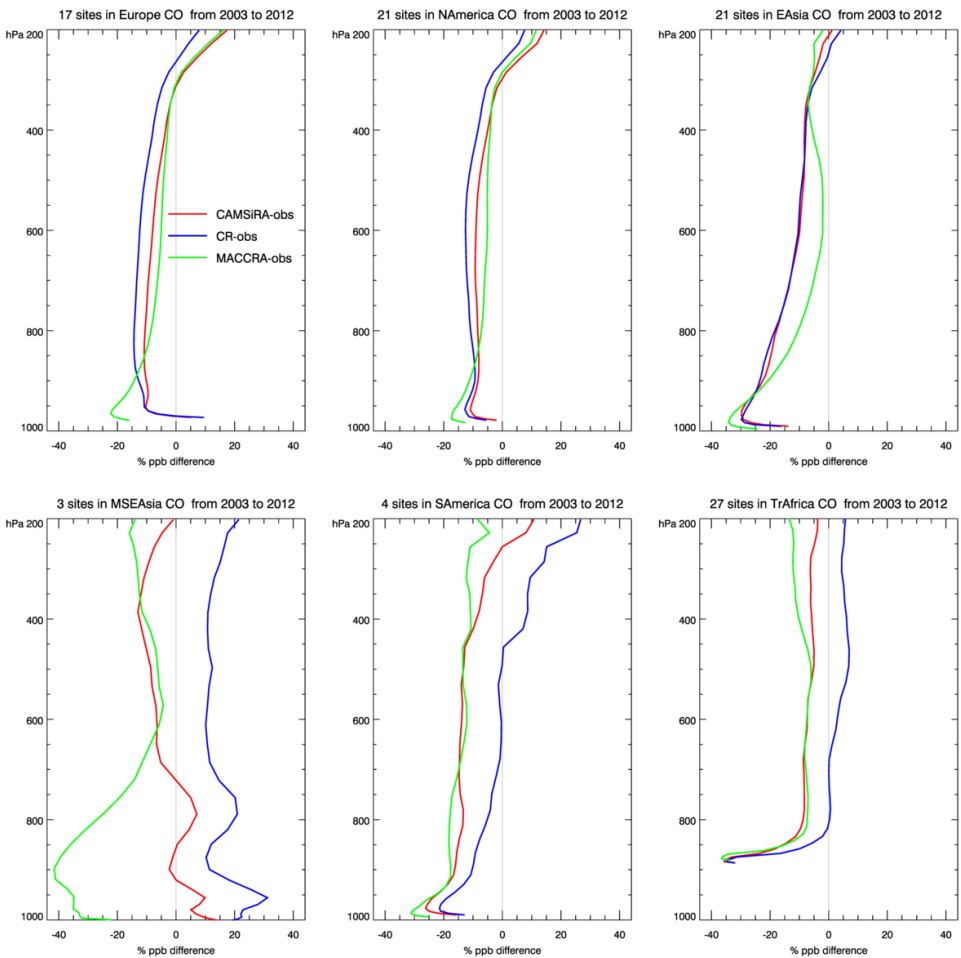

*Figure 5 Average relative bias (%) in CO of CAMSiRA, MACCRA and CR against MOZAIC / IGAOS flight profiles averaged over different regions (see Table 3) for the period 2003–2012.*






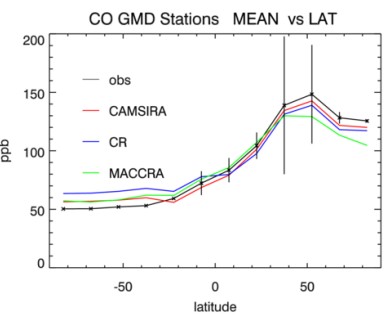 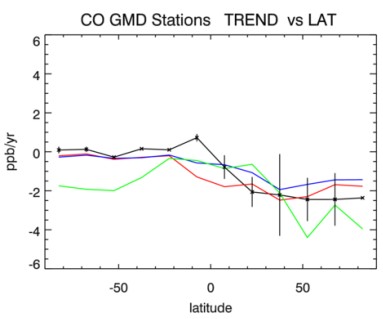


*Figure 6 Zonal average of mean surface CO in ppb observed at NOAA-GMD stations*
*(2003–2014) and values from CAMSiRA, CR and MACCRA (2003–2012) (left) and zonal*
*median of linear trend in ppb/yr (right). The error bars indicate the range of the observed*
*values.*







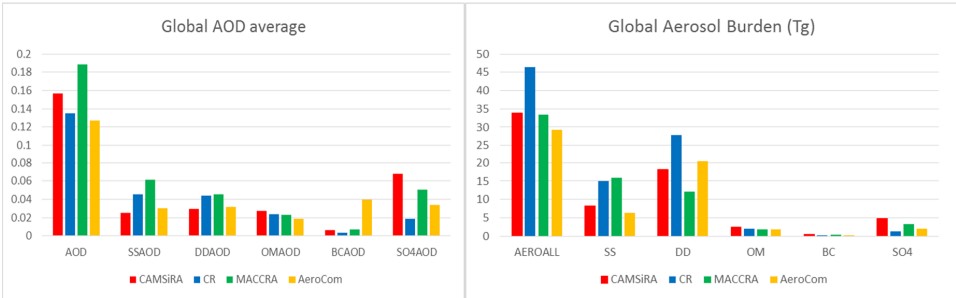


*Figure 7 Globally average of total AOD (550 nm) and species AOD (left) and global total*
*and species and burden in Tg (right) of sea salt (SS), desert dust (DD), organic matter*
*(OM), black carbon (BC) and sulphate aerosol (SO₄) for CAMSiRA (red\), CR (blue) and*
*MACCRA (green) and the median of the AeroCom model inter-comparison (yellow, Kinne*
*et al., 2006 and Textor et al., 2006).*





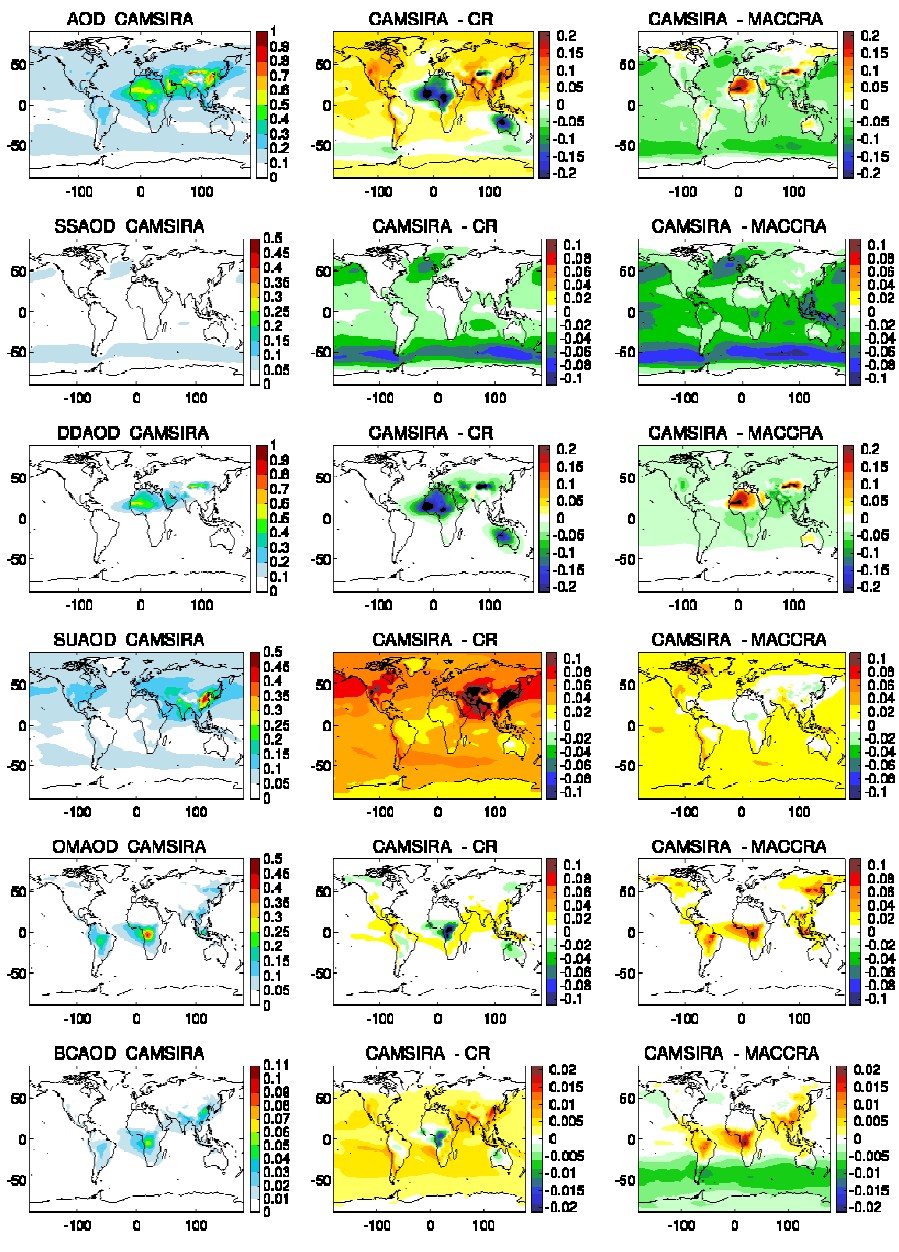


*Figure 8 Total average AOD (row 1, scale max 1.0), AOD of desert dust (row 2, 1.0), sea*
*salt (row 3, 0.5), sulphate (row 4, 0.5), organic matter (row 5, 0.5) and black carbon (row*
*6, 0.11) of CAMSiRA (average 2003–2015, left) and differences against CR (average 2003–*
*2015, middle) and MACCRA (average 2003–2012, right).*





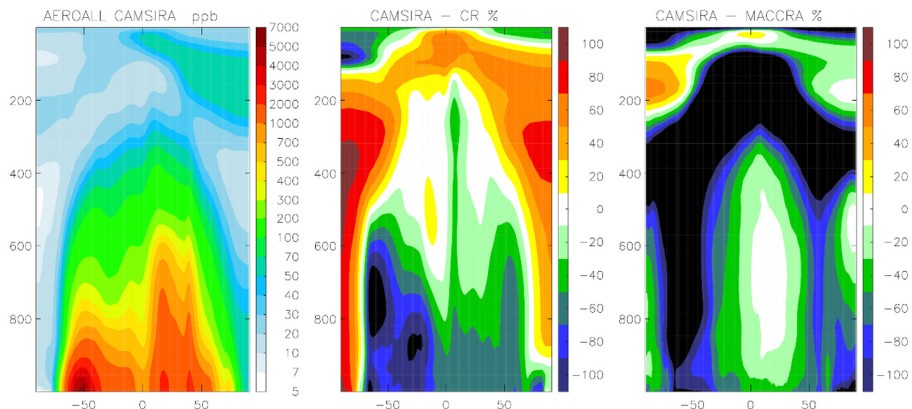


*Figure 9 Zonally averaged total aerosol mass mixing ratio ($10^{-9}$kg/kg) of CAMSiRA (2003–*
*2015, left) and relative difference (%) against CR (2003–2015, middle) and MACCRA*
*(2003–2012, right).*





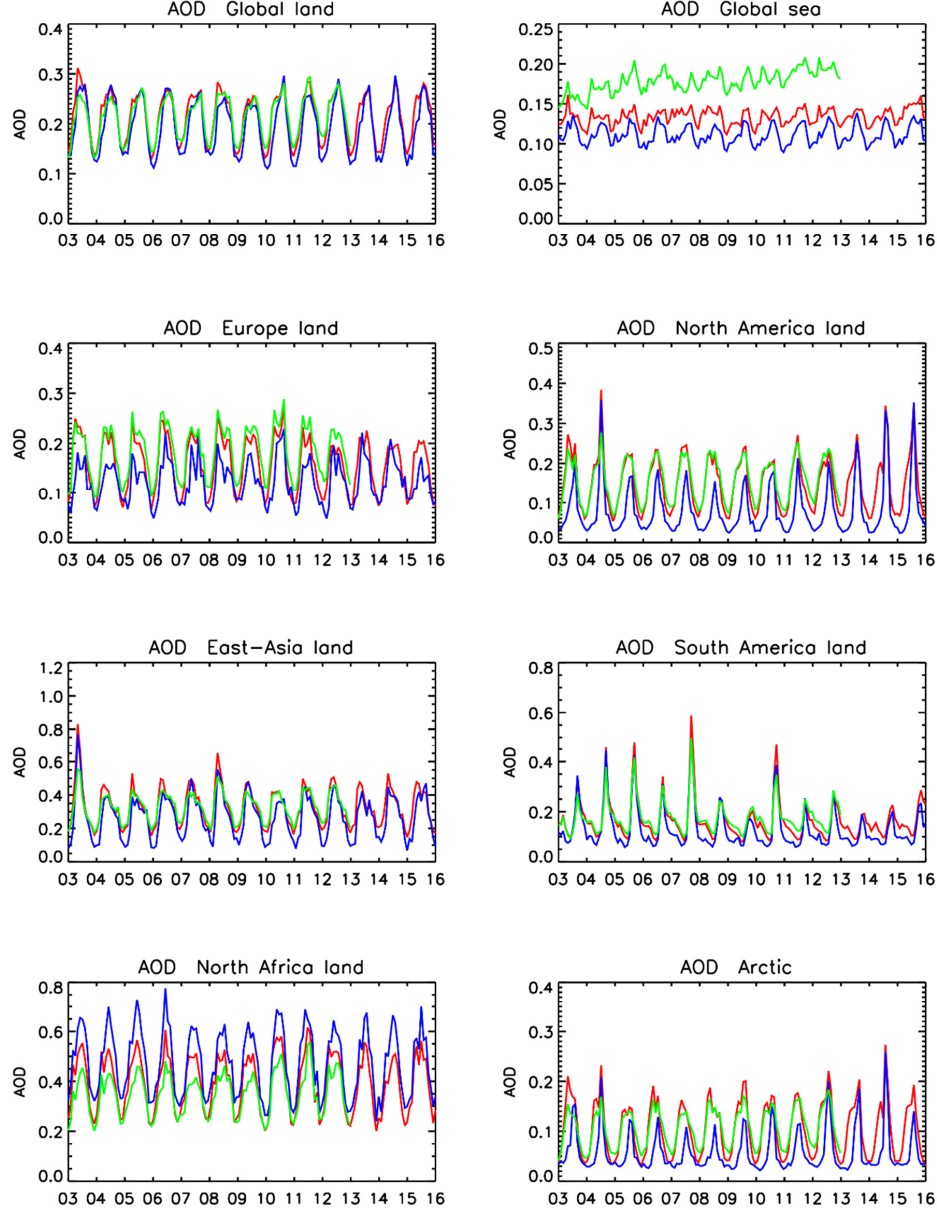


*Figure 10 Time series of monthly mean AOD over the whole globe (land or seas points)*
*and for different regions (see Table 3) for the period 2003–2015 from CAMSiRA (red), CR*
*(blue) and MACCRA (green, 2003–2012).*






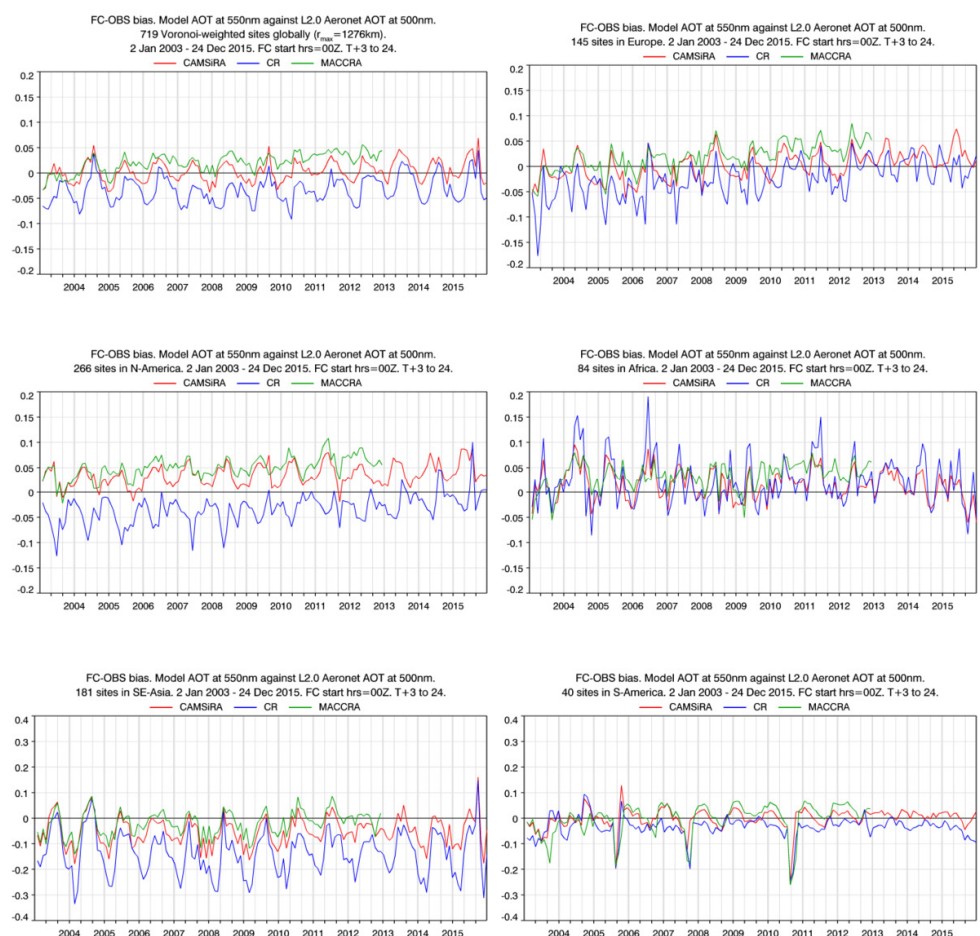


*Figure 11 Time series of monthly mean bias against AERONET AOD observations*
*averaged over the whole globe (top left), Europe (top right), North America (middle left),*
*Africa (middle right), South East Asia (bottom left) and South America (bottom right) for*
*CAMSiRA (red), CR (blue) and MACCRA (green).*





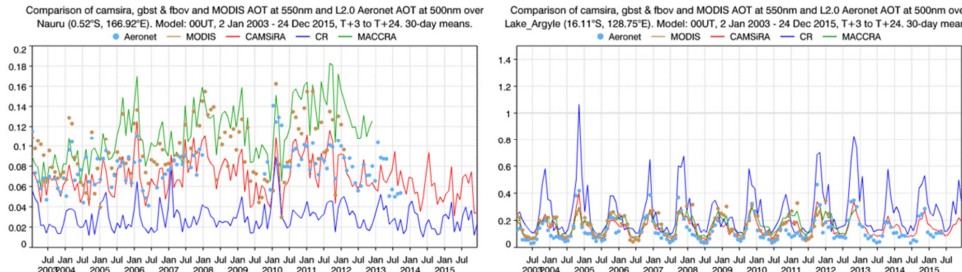


*Figure 12 Time series of monthly mean AOD from AERONET observations (light blue*
*dots), MODIS retrievals (brown dots) and from CAMSiRA (red), CR (blue) and MACCRA*
*(green) at Nauru (left) and Lake Argyle (right).*




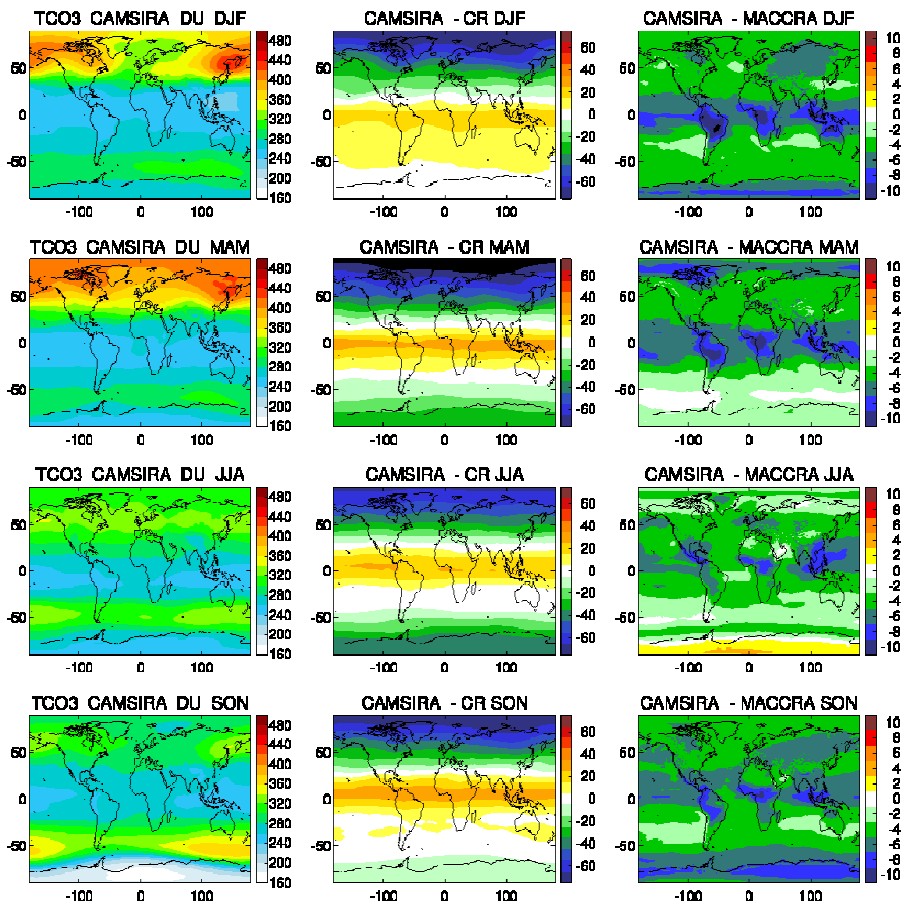


*Figure 13 Seasonal averaged TC ozone (DU) from CAMSiRA (left), difference between
CAMSiRA and CR (middle) and CAMSiRA and MACCRA (right, 2003–2012, different
scale) for the seasons DJF (row 1) MAM (row 2), JJA (row 3) and SON (row 4).*



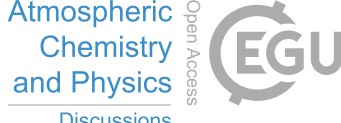

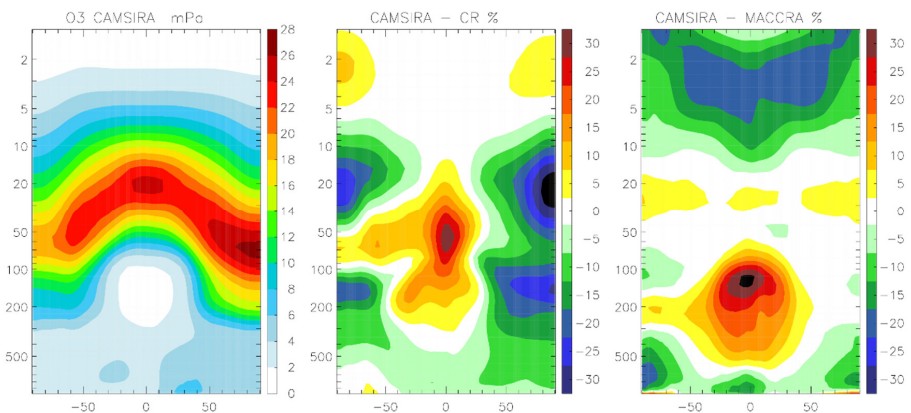


*Figure 14 Zonally averaged ozone partial pressure (mPa) of CAMSiRA (2003–2015, left)*
*and relative difference (%) against CR (2003–2015, middle) and MACCRA (2003–2012)*





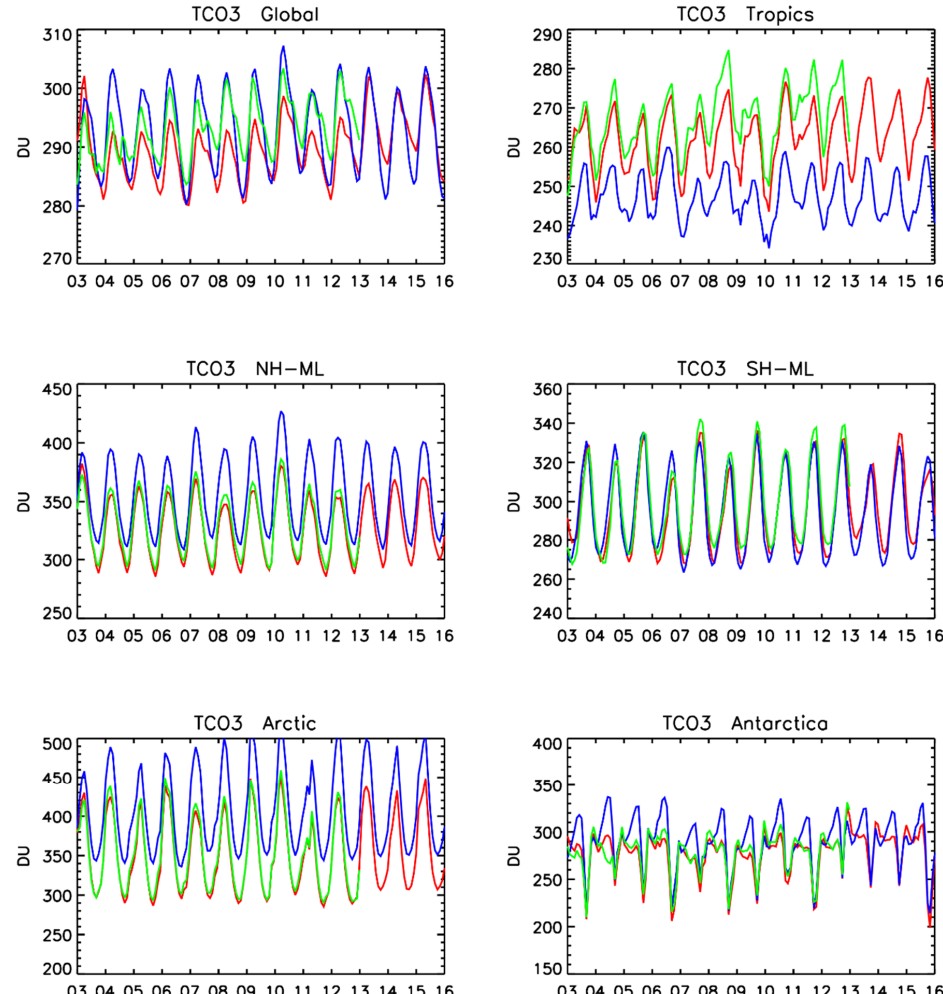


*Figure 15 Monthly ozone TC (DU) area averaged over different regions (see Table 3) from*
*CAMSiRA (black), CR (blue) and MACCRA (green) for 2003–2015.*




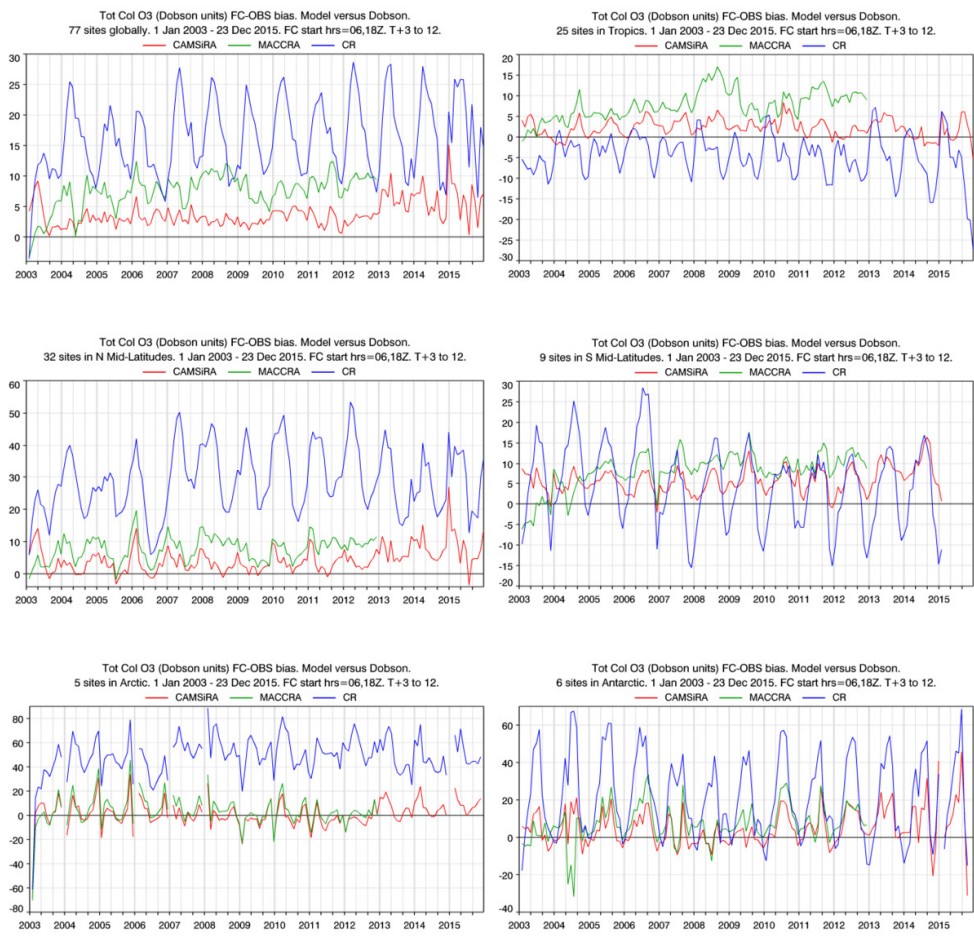


*Figure 16 Time series of monthly mean bias in DU against WOUDC Dobson sun*
*photometers for the globe (top left), the tropics (top right), NH mid-latitudes (middle left),*
*SH mid-latitudes (middle right), the Arctic (bottom left) and Antarctica (bottom right) for*
*CAMSiRA (red), CR (blue) and MACCRA (green).*





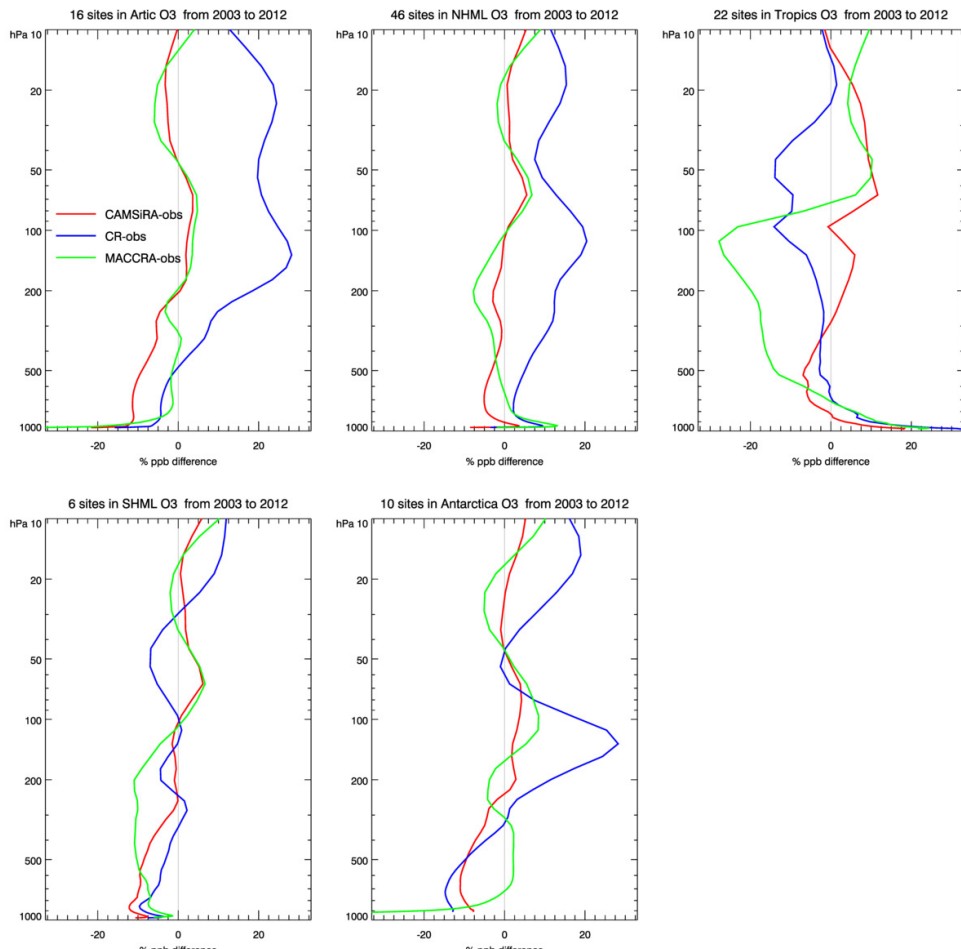


*Figure 17 Mean relative bias of CAMSiRA (red), MACCRA (green) and CR (blue) against*
*ozone sondes in the Arctic (top left), NH mid-latitudes (top middle), Tropics (top right),*
*SH-mod-latitudes (bottom left) and Antarctica (bottom middle) for the period 2003–2012.*





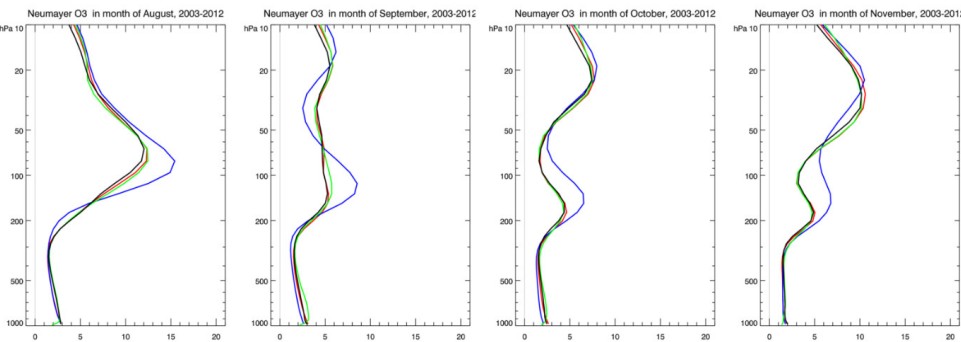


*Figure 18 Monthly mean ozone profiles (mPa) at Neumayer Station from ozone sondes, of*
*CAMSiRA (red), MACCRA (green) and CR (blue) for August to November (2003–20012).*





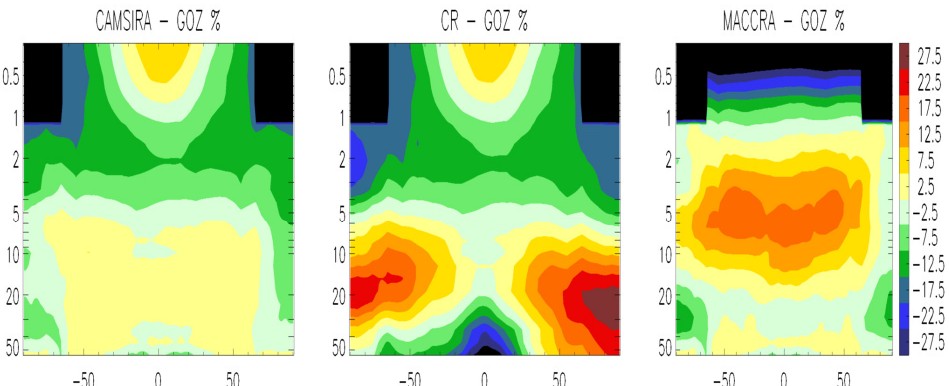


*Figure 19 Cross sections (50–0.3 hPa) of the relative biases of zonally averaged ozone (%)*
*of CAMSiRA (left), CR (middle) and MACCRA (right) against the GOZCARDS product*
*(GOZ) for the period 2005–2012.*





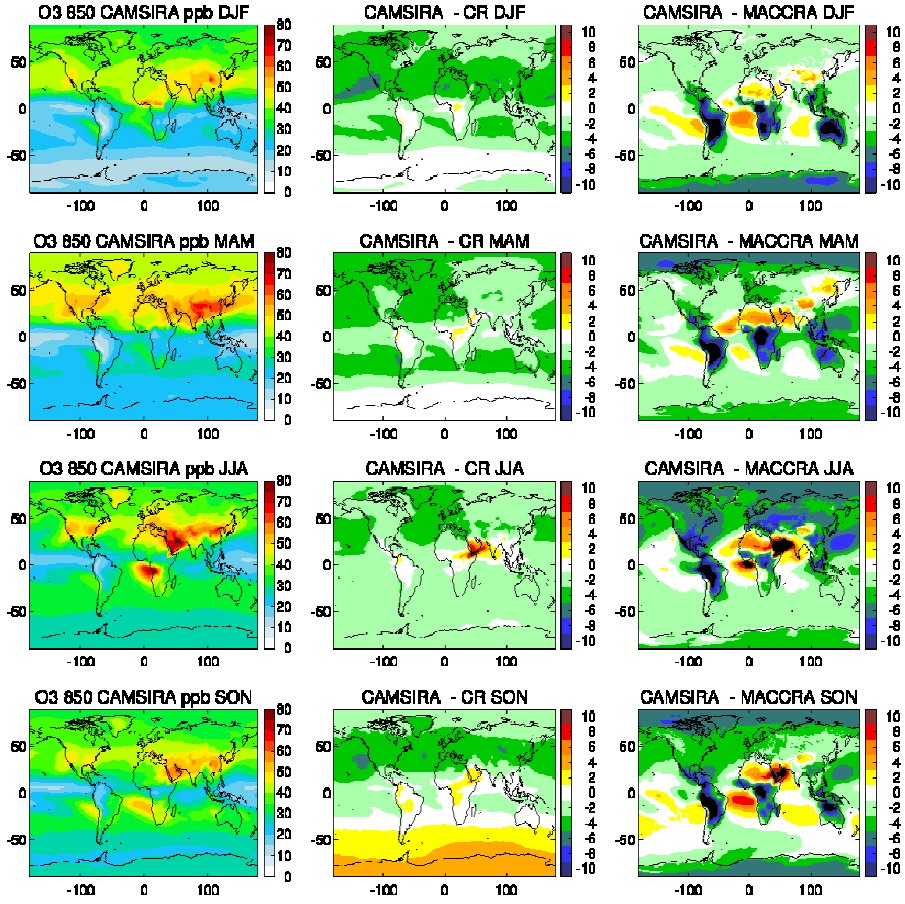


*Figure 20 Seasonal averaged ozone at 850 hPa (ppb) from CAMSiRA (left), difference*
*between CAMSIRA and CR (middle) and CAMSiRA and MACCRA (right, 2003–2012) for*
*the season DJF (row 1), MAM (row 2), JJA (row 3) and SON (row 4).*





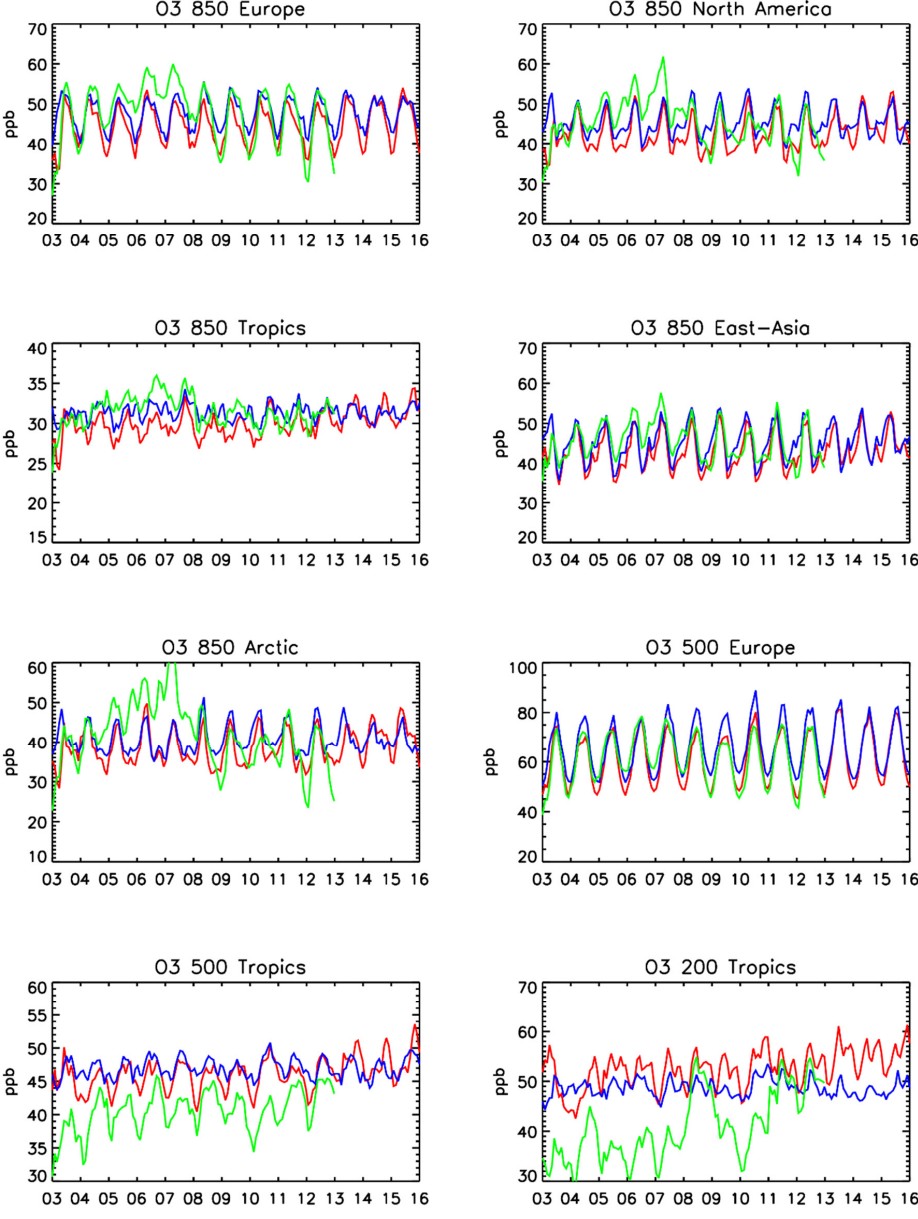


*Figure 21 Monthly ozone volume mixing ratio at 850, 500 and 200 hPa over different*
*regions (see Table 3) from CAMSiRA (red), CR (blue) and MACCRA (green) for 2003–*
*2015.*






*Figure 22 Time series of seasonal mean ozone bias in ppb in the pressure ranges 950-700,*
*700-400 and 400-300 hPa against ozone sondes at Ny-Ålesund, DeBilt, Huntsville, Hong*
*Kong Observatory, Nairobi and Neymayer station for CAMSiRA (red), CR (blue) and*
*MACCRA (green).*





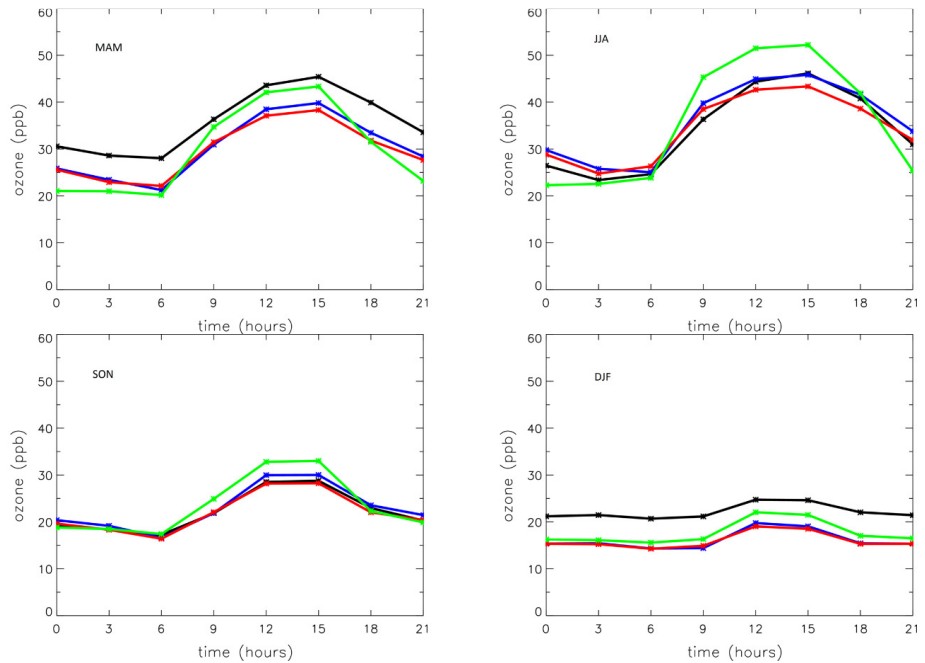


*Figure 23 Average diurnal cycle of ozone at EMEP-AirBase stations in Europe (black) for*
*the seasons MAM (top left), JJA (top right), SON (bottom left) and DJF (bottom right) for*
*CAMSiRA (red), CR (blue) and MACCRA (green).*





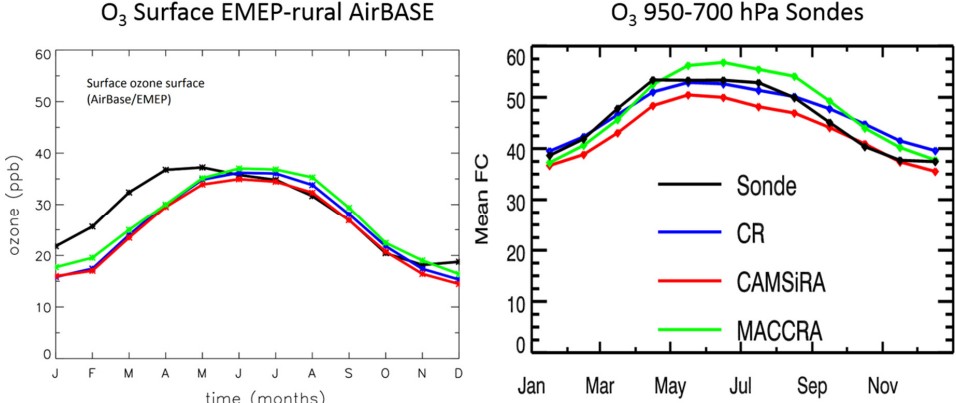


*Figure 24 Average seasonal cycle of surface ozone at EMEP-AirBase stations (left) and at*
*European ozone sonde sites in the pressure range (950–700 hPa) for CAMSiRA (red), CR*
*(blue) and MACCRA (green).*