# Peer review of "The CAMS interim Reanalysis of Carbon Monoxide, Ozone"

_Atmospheric Chemistry and Physics, 2016_

## Referee Comment (RC1) · Anonymous Referee #1 · 30 Sep 2016

I think this is a standard paper describing a reanalysis product. It is likely to be useful to the scientific community; however, it is a bit difficult to read, partly owing to its length. The paper should be accepted for publication once the authors address a series of points, detailed in the specific comments and technical comments. They largely concern quantification and/or clarification of statements made.

Specific comments:

L. 32: Indicate why ozone at the surface cannot be improved by the assimilation.

L. 56: Quantify the "sufficient accuracy".

L. 58: Provide details of the surface properties.

L. 109: List the key species.

L. 208: Why did you use scenario 8.5 instead of another one?

L. 286-287: Did you use the averaging kernels for data other than MOPITT? Explain your choices.

L. 291: The data used are flagged "good" or not flagged "bad"?

L. 375: Does the decrease in the burden indicate a positive result from the assimilation?

L. 397: Explain in the text why you do not assimilate MOPITT observations over the Arctic.

L. 451: Why is there only a little effect on the surface? Why are there no changes between CR and CAMSiRa from the assimilation?

L. 471: Is it reasonable to calculate a linear trend? What assumptions do you make?

L. 516: Provide references for this statement.

L. 523: Is the comparison with MACCRA and CAMSiRA within the errors of these datasets?

L. 535: Why is there an exaggeration of the sea salt emission?

L. 594: Discuss why this seems unrealistic.

L. 662: Quantify the trends. Explain (or remind the reader) how you test for significance. Same for L. 751 and L. 757.

L. 674: Provide further details of the artificial accumulation of sulphate by the assimilation.

L. 767: Why is this remarkable? Because unexpected? Please avoid subjective comments.

L. 852-865: What is the fidelity of the GOZCARDS dataset?

Technical comments:

L. 36: Do you need "clearly"?

L. 128: practise -> practice.

L. 159: were -> are.

L. 167: I suggest you do not start a sentence with an acronym.

L. 221: Have you introduced the acronym for POET?

L. 389: "in the" repeated.

L. 535: Replace "probably" by "likely". Do this elsewhere as well.

L. 551: was -> were.

L. 799: "…V3.4) is at the…"

L. 835: latitudes.

L. 978: CAMSiRA.

---

## Referee Comment (RC2) · Anonymous Referee #2 · 1 Oct 2016

This manuscript presents a thirteen-year reanalysis of Carbon Monoxide, Ozone and Aerosol obtained from the CAMS system. Satellite measurements were assimilated using the 4D-VAR technique. The obtained data sets were validated against various independent observations, and then the validation results were compared with those of the previous MACC reanalysis and the control experiment. Compared to the previous MACC reanalysis, there were clear improvements in the CAMS system for most cases, owing to the improved data assimilation system including the improved treatment of observational data and improved emission setting. The manuscript is generally well written, and I believe it can contribute to various studies. As this study represents the first trial reanalysis of trace gases and aerosols simultaneously, I was expecting more discussions on the synergetic effect. I recommend this manuscript to be accepted for publication after a few revisions.

[Figure]

Major comments;

1. An important advantage of this study is the simultaneous analysis of trace gases and aerosols within the same data assimilation framework. However, it is unfortunate that their interactions were not considered in the current setting. More discussions on their potential would still be useful. I suggest discussing this topic in an additional section, for instance, how much changes in trace gas concentration can be expected using the analyzed aerosol fields, and if these changes bring further improvements in the trace gas analysis (and vice versa). Although the paper is already very long, presenting several sensitivity calculation results could be helpful.

2. As the system was developed at meteorological operational centers, the authors may want to discuss more about the contribution of the CAMS interim reanalysis to meteorological and climate activities. This discussion would be useful to many readers in understanding how the composition and aerosol reanalysis will be helpful in wide research fields.

3. In Section 3, the differences in CO between the systems are primarily explained by surface emissions. There could also be clear differences in OH and natural CO sources by oxidation, which may explain the CO differences. Further discussions may be required on the estimated CO error difference.

Specific comments:

L. 394: "Owing to the hemispheric...". This sentence is not clear.

L.404-407: It is surprising that, even after correcting the concentration by data assimilation, the influence of different emission data is so large. Does this mean that the observational constraints are insufficient to remove the influence of a priori model errors? Further discussions would be helpful.

L. 437-440: How does the bias vary with year?

L. 798-799: "However, the change of..." This sentence is not clear.

L. 1008-1010: How long was the spin-up period?

---

## Author Response (AR1)

**2 Response to Review #1**

We thank reviewer #1 for the review, the insightful comments to the paper and for his/herendurance to read the long paper.

5 We will respond to the review point by point. The reviewer's comments are included in bold6 italics.

**I think this is a standard paper describing a reanalysis product. It is likely to be useful to the scientific community; however, it is a bit difficult to read, partly owing to its length.**

9 We appreciate the reviewer's concern that the paper is long. However, we prefer to present 10 all considered species in one paper because they were produced by the CAMS system in one 11 combined assimilation experiment.

12 The paper should be accepted for publication once the authors address a series of points, 13 detailed in the specific comments and technical comments. They largely concern 14 quantification and/or clarification of statements made.

15 Specific comments:

**16 L. 32: Indicate why ozone at the surface cannot be improved by the assimilation.**

17 The surface values could potentially be changed by the following processes: (i) direct addition 18 of observation increments close to the surface, (ii) the impact of non-surface observations or 19 the observation of other species by means of the model backward error co-variances and (iii) 20 the downward transport of ozone from levels where the assimilation changed the ozone fields.

We think that the impact of any of these three factors was eliminated at the lowest modellevels by the strength of the ozone dry deposition and titration with NO near the surface.

CAMSiRA did not assimilate any surface observations nor satellite retrievals for the lower troposphere. Only total columns and stratospheric profile data were assimilated. The background error co-variances calculated with the NMC method did not provide enough impact for strong non-local vertical influence, which would have led to an alteration at surface. Also, species-to-species back-ground error correlation were not implemented in the applied 4D-VAR method.

- We added in the abstract:
- 30 " ... because of the strong impact of surface processes such as dry deposition and titration
- 31 with nitrogen monoxide (NO), which were both not changed by the assimilation. "
- We add in the conclusion section when we discussed the reasons for the small influence(L1147)
- 34 "... nor that the vertical correlations in the model background errors were strong enough to35 cause a correction of the surface levels based on the levels above. "
- 36 L. 56: Quantify the "sufficient accuracy".
- 37 We can not quantify this rather general statement but we changed it as follows:
- 38 ".. with an accuracy sufficient to have an impact during the assimilation. "
- 39 L. 58: Provide details of the surface properties.
- 40 We clarified the statement by replacing "surface properties" with "surface albedo".
- 41 L. 109: List the key species.
- 42 We changed the text as follows:
- 43 "... the aerosol variables and key chemical species such as ozone, HNO3, N2O5, NO, NO2,
  44 PAN and SO2 only".

**45 L. 208: Why did you use scenario 8.5 instead of another one?**

- 46 The scenario was chosen by the producers of the MACCITY data set (Granier et al. ,2011)
- 47 We added "... obtained in the MACCity emissions ..."

**48 L. 286-287: Did you use the averaging kernels for data other than MOPITT? Explain your 49 choices.**

- 50 We add
- 51 "For the ozone retrieval averaging kernels were not used because they were not provided or 52 did not improve the analysis. For example, the high vertical resolution of the MLS ozone
- 53 retrievals in the stratosphere made the use of AK not necessary. "
- 54
- 55 The L. 291: The data used are flagged "good" or not flagged "bad"?

- 56 Yes, this is the case. The retrieval data include a quality flag given by the providers
- 57 L. 375: Does the decrease in the burden indicate a positive result from the assimilation?
- 58 Yes, because there is a better agreement with the MOPITT total column retrievals.

**59 L. 397: Explain in the text why you do not assimilate MOPITT observations over the Arctic.**

- 60 Larger biases and errors in the retrievals occur at high latitudes because of low thermal
- 61 contrast. It is a recommendation by the data providers not to assimilate data in high latitudes.
- 62 We added: "... because of the higher biases of the MOPITT data in this region."

**L. 451: Why is there only a little effect on the surface? Why are there no changes between CR and CAMSiRa from the assimilation?**

65 See our response above

**66 L. 471: Is it reasonable to calculate a linear trend? What assumptions do you make?**

It is a valid to comment to question the underlying assumption (i.e. a linearity) for any type of trend analysis. A detailed trend analysis is beyond the scope of the paper. However, the linearity of the trend seems not an unreasonable choice when looking at the graphs. Our focus is the comparison of trends of different data sets using a unified but simple approach.

- 71 We will add:
- 72 "... and, for reasons of simplicity, only the linear ..."

We also point out that linear trends are often expressed in units of %/yr in the paper. We concede that this unit is technically not consistent with a linear trend. We obtain the liner trend as percentage by normalising the linear trend (e.g. Tg/year) with the average of the quantity over the whole period, i.e. all years.

- We add at line 361
- 78 "The linear trend is as expressed as percentage with respect to the mean of the burden over the79 whole period."
- 80 L. 516: Provide references for this statement.
- 81 We added the following reference:
- 82 (Eskes et al., 2015).

- 83 Eskes, H., Huijnen, V., Arola, A., Benedictow, A., Blechschmidt, A.-M., Botek, E., Boucher,
- 84 O., Bouarar, I., Chabrillat, S., Cuevas, E., Engelen, R., Flentje, H., Gaudel, A., Griesfeller, J.,
- 85 Jones, L., Kapsomenakis, J., Katragkou, E., Kinne, S., Langerock, B., Razinger, M., Richter,
- 86 A., Schultz, M., Schulz, M., Sudarchikova, N., Thouret, V., Vrekoussis, M., Wagner, A., and
- 87 Zerefos, C.: Validation of reactive gases and aerosols in the MACC global analysis and
- 88 forecast system, Geosci. Model Dev., 8, 3523-3543, doi:10.5194/gmd-8-3523-2015, 2015.
- 89

**90 L. 523: Is the comparison with MACCRA and CAMSiRA within the errors of these 91 datasets?**

92 Unfortunately, we cannot provide error estimates of the global burden of the two analysis sets.

**93 L. 535: Why is there an exaggeration of the sea salt emission?**

- 94 As pointed out in the supplement, sea salt emissions were close to the median of the Aerocom
- 95 models. They were only at the high end of the values given in Boucher et al. (2015)
- 96 We amend the text as follows:
- 97 "The simulated sea salt emissions of C-IFS were within the reported range in the literature
- 98 (see supplement). This suggests that the loss processes of sea salt were underestimated in C-
- 99 IFS in comparison to other models."

**100 L. 594: Discuss why this seems unrealistic.**

- 101 We add the following
- 102 "..., given that the global SO2 emission are only less than 2% of the total aerosol emissions103 (see supplement)"

**104 L. 662: Quantify the trends. Explain (or remind the reader) how you test for significance. 105 Same for L. 751 and L. 757.**

- 106 The significance of the linear trends was estimated at the 95 confidence interval. We now107 repeat this information in each section.
- 108 L. 674: Provide further details of the artificial accumulation of sulphate by the 109 assimilation.
- 110 We added:

- 111 "The increase in sulphate was probably caused by underestimated loss processes for sulphate
- and  $SO_2$  in the free and upper troposphere away from the emissions sources. The relative increase in sulphate with respect to the other aerosol species could not be corrected by the assimilation of AOD."

**115 L. 767: Why is this remarkable? Because unexpected? Please avoid subjective comments.**

- 116 We replace the statement with:
- 117 "Despite its simplicity, the Cariolle scheme in CR reproduced the..."

**118 L. 852-865: What is the fidelity of the GOZCARDS dataset?**

The standard error of the GOZCARDS data set is given as part of the data set. The values of the error are in the range of 10-20 ppb on the considered region, which is about 1%. However this error does not reflect biases. As we already mention in the text, the inter-comparison of different satellite retrievals by Tegtmeier et al. (2013) shows that MLS is biased low above 5 hPa (5-10%) and ACE-FTS is biased high above 10 hPa and biased low below 10 hPa with respect to the multi-instrument-mean. Since ACE-FTS contributes more to the GOZCARDS product in this region, we assume that the GOZCAD biases are controlled by the ACE-FTS

- 126 biases.
- 127 We will quantify the biases in the text:
- 128 "ACE-FTS is biased high (5-10%) above 10 hPa and biased low (5-10%) below 10 hPa
- 129 against the median of various retrievals. "
- 130 We corrected all technical comments.
- 131

132

**133 *Response to Review #2**

134

We thank reviewer #2 for the review, the insightful comments to the paper and for his/herendurance to read the long paper.

137 We will respond to the review point by point. The reviewer's comments are included in bold138 italics

**139 Major comments;**

140 1. An important advantage of this study is the simultaneous analysis of trace gases and aerosols within the same data assimilation framework. However, it is unfortunate that their 141 interactions were not considered in the current setting. More discussions on their potential 142 143 would still be useful. I suggest discussing this topic in an additional section, for instance, 144 how much changes in trace gas concentration can be expected using the analyzed aerosol 145 fields, and if these changes bring further improvements in the trace gas analysis (and vice versa). Although the paper is already very long, presenting several sensitivity calculation 146 147 results could be helpful.

We fully agree that interactions between chemistry and aerosol within a data assimilation frame work is an important topic. However, its study will be more the focus of ongoing and future work and it is not the result of the work presented here. In the current version of the manuscript we mention the prospects in the conclusion section (L 1155).

In the CAMSiRA (and MACCRA) no interaction between aerosol, chemistry and meteorology was simulated. The only potential interaction would be the impact of the tropospheric ozone assimilation on CO and vice versa. As reported in Inness et al. (2015) the applied system does not show a strong inter-species synergy, in particular as no NO2 retrievals were assimilated in CAMSiRA. A further explanation for the lack of synergies is that no adjoint and tangent linear formulation of the chemistry scheme was applied and that no species-to-specie background error covariances were considered in our 4D-VAR approach.

In the next version of the CAMS system, the impact on assimilated aerosols and ozone in the radiation scheme, the impact of aerosol on photolysis rates and on some heterogeneous reaction (N2O5, HO2) will be considered. 162 To clarify that the assimilation system used for the paper does not present these interactions to 163 a large extend, we add the following statement in section 2.4 (C-IFS data assimilation).

"A further potential interaction between the assimilated species could be introduced by the adjoint and tangent linear representations of the chemical mechanism and the aerosol module as part of the 4D-VAR approach. The applied tangent linear and adjoin formulation of C-IFS accounts only for transport processes and not the sources and sinks of atmospheric composition. Because of this limitation and the lack of aerosol-chemistry-meteorology feedbacks in the C-IFS version used in this study, interactions among species and with the meteorology as part of the assimilation are not represented in CAMSiRA. "

171

172 2. As the system was developed at meteorological operational centers, the authors may want 173 to discuss more about the contribution of the CAMS interim reanalysis to meteorological 174 and climate activities. This discussion would be useful to many readers in understanding 175 how the composition and aerosol reanalysis will be helpful in wide research fields.

176 In the current version of the manuscript we mention applications of the re-analysis of 177 atmospheric composition, such as boundary condition for regional models and trace-gas 178 climatologies in the introduction and in the conclusions.

179 We can report that new trace-gas climatologies for ozone and aerosol were compiled from 180 CAMSiRA and implemented in the new cycle of the operational ECMWF NWP model. In 181 particular the reduced ozone bias in the upper stratosphere and mesosphere led to an improved 182 skill forecasts this See in temperature in region. 183 https://software.ecmwf.int/wiki/display/FCST/Implementation+of+IFS+Cycle+43r1

184 A report/paper is in preparation but not yet ready to be cited.

An other application of CAMSiRA is the analysis of trends, which we demonstrate on the example of CO surface data (Figure 6). Finally, the evaluation of model runs would be a new application for AC re-analysis data. However, we would leave it (within the scope of the paper) to the reader to decide if there is enough confidence that CAMSiRA is well suited for this purpose.

190 3. In Section 3, the differences in CO between the systems are primarily explained by

surface emissions. There could also be clear differences in OH and natural CO sources by

192 *oxidation, which may explain the CO differences.*

When discussing the global patterns of the differences between CAMSiRA and CR we actually come to the conclusion (L 337) that "... global chemical loss and production of CO as well as problems with the large scale transport. ... " and less the CO emissions itself are the reason for the biases of the model.

197 We find it difficult to distinguish with the discussed model runs to clarify in detail if 198 emissions and distribution of CO pre-courser species such as VOCs and CH4 or a reduced CO

199 lifetime because of higher OH values are more likely the reasons for the identified CO biases.

200 In any case we conclude that the CO emissions are not the sole reason for the CO biases.

201 We mention this in the conclusion of the paper (L 1128) but will refine the statement to:

202 "However, the rather zonally homogeneous CO differences between CR and CAMSiRA
203 suggest that not only biases in the fire emissions but also of the CO lifetime and chemical
204 production as well as the CO transport need to be investigated further. "

205

**206 Specific comments:**

**207 L. 394: "Owing to the hemispheric...". This sentence is not clear.**

208 We reformulate as follows:

209 "Because of the hemispheric influence, i.e. the hemispheric reduction in CO, the CO trend in210 CR over Eastern China became negative in the middle troposphere."

211

L.404-407: It is surprising that, even after correcting the concentration by data assimilation, the influence of different emission data is so large. Does this mean that the observational constraints are insufficient to remove the influence of a priori model errors? Further discussions would be helpful.

In the current approach the surface emissions are not changed by the assimilation of CO
observations. This has been identified as a topic for future developments in the conclusions
(see L 1185).

The missing total agreement with the observations at the time of the analysis is also caused by the relative size of the observation and background error statistic. The background error for CO is calculated using an ensemble of forecasts, which only accounts for the variability in the

- transport (winds) and not for the uncertainty of the emissions. The background error at the
- surface is therefore most likely underestimated leading to an "over-confidence" in the model
- as part of the assimilation process.
- 225 We will add at line (L 274, Section C-IFS data assimilation)

226 "However, the ensemble did not account for the uncertainty of the emissions, which leads to 227 an underestimation of the background error for CO."

And we will ad in the section on recommendations (L 1187)

"A promising development is to enable the correction of emissions with the C-IFS data assimilation system based on observations of atmospheric composition. This could also improve the analysis of tropospheric ozone as ozone precursor emissions would be corrected. An intermediate step in this direction is to better account for the emission uncertainty in the model background error statistics. "

**234 L. 437-440: How does the bias vary with year?**

The data coverage of the MOZAIC/IAGOS data varies a lot so that a robust conclusion for the year-to-year variability would be difficult to obtain. However, we discuss the agreement of the trends for surface CO observations in section 3.4. and show a good correspondents in the observed, modelled (CR) and assimilated trends (CAMSiRA).

A conclusion of the discussion of the inter-annual variability of the CO burdens (section 3.2)
over Europe and North-America is that there is better agreement between CR and CAMSiRA
at the end of the period. This could indicate that the biases of the anthropogenic emissions

242 decrease from 2003 to 2015.

**243 L. 798-799: "However, the change of..." This sentence is not clear**

244 We reformulate as follows:

- "It is not caused by the change of the assimilated MLS version (from V2 to V3.4) because thistook place already at the beginning of 2013 (see Table 2)."
- 247
- 248 L. 1008-1010: How long was the spin-up period?
- 249 The MACCRA was started on the 1.12.2002 (Inness et al. 2013)
- 250 We add " ... and the short spin-up period of only 1 month"

254

**The CAMS interim Reanalysis of Carbon Monoxide, Ozone and Aerosol for 2003–2015**

- J. Flemming1, A. Benedetti1, A. Inness1, R. Engelen1, L. Jones1, V. Huijnen2, S.
- Remy3, M. Parrington1, M. Suttie1, A. Bozzo1, V.-H. Peuch1, D. Akritidis4 and E.
  Katragkou4

[revised manuscript text omitted]

|--------------------|--------------------------|--------------------|-----------------------|--------------|-----------------|
| MOPITT             | Deeter et al. (2011)     | V5 TIR             | 20030101-             | CO TC        | 65N-65S         |
| Terra              |                          | NRT                | 20121218              |              | QC=0            |
| CONT               | M                        |                    | From 20121219         | 02           | 20NI 20C        |
| GOME               | Munro et al. (1998)      |                    | 20030101-
20030531 | O3 profile   | 80N-80S         |
| EKS-2              |                          |                    |                       |              | SOE>15,
QC=0 |
| GOME-2             | Hao et al. (2014)        | NRT                | 20120901-             | O3 TC        | SOE>10          |
| Metop A            |                          | GDP4.4             | 20130714              |              | QC=0            |
|                    |                          | NRT
GDP4.7      | From 20130715         |              |                 |
| GOME-2             | Hao et al. (2014)        | NRT                | From 20140101         | O3 TC        | SOE>10          |
| Metop B            |                          | GDP4.7             |                       |              | QC=0            |
| MIPAS              | von Clarmann et al.      | NRT                | 20030101-             | O3 profile   | QC=0            |
| Envisat            | (2003, 2009)             | CCI                | 20040326              |              |                 |
|                    |                          |                    | 20030127-
20120331 |              |                 |
| MLS                | Froidevaux et al. (2008) | V2                 | 20040808-             | O3 profile   | QC=0            |
| Aura               |                          | NRT V3.4           | 20121231              |              |                 |
| OMI                | $L_{in}$ at al. (2010)   | V002               | 20041001              | 02 TC        |                 |
|                    | Liu et al. (2010)        | NRT                | 20121231              | 0510         | 30E > 10        |
| Лша                |                          | INKI               | From 20130101         |              | QC-0            |
| SBUV/2 NOAA-       | Bhartia et al. (1996)    | V8                 | 20040101-             | O3 PC        | SOE>6           |
| 16                 |                          |                    | 20081020              | 6 layers     | QC=0            |
| SBUV/2 NOAA-       | Bhartia et al. (1996)    | V8                 | 20030101-             | O3 PC        | SOE>6           |
| 17                 |                          |                    | 20121130              | 6 layers     | QC=0            |
| SBUV/2 NOAA-
18 | Bhartia et al. (1996)    | V8                 | 20050604–
20121217 | O3 PC        | SOE>6           |
|                    |                          |                    |                       | 6 layers     | QC=0            |
| SBUV/2 NOAA-       | Bhartia et al. (1996)    | V8                 | From 20090100         | O3 PC        | SOE>6           |
| 19                 |                          |                    |                       | 6 layers     | QC=0            |
| SCIAMACHY          | Eskes et al. (2012)      | CCI                | 20030101-             | O3 TC        | SOE>6           |
| Envisat            |                          |                    | 20120408              |              | QC=0            |
| MODIS / Terra      | Remer et al. (2005)      | Col.5              | 20030101-             | AOD
550nm | 70N-70S         |
|                    |                          | NRT Col.5          | From 20080801         | 5501111      |                 |
| MODIS / Aqua       | Remer et al. (2005)      | Col.5
NRT Col.5 | 20030101-             |              | 70N-70S         |
| ino 210 / riqua    |                          |                    | 20080731              | 550nm        | /011 /00        |
|                    |                          |                    | From 20080801         |              |                 |

Table 2 Assimilated satellite observations in CAMSiRA

| Area                     | Coordinates           |  |  |
|--------------------------|-----------------------|--|--|
| North America            | 165°W-55°W, 25°N–75°N |  |  |
| Europe                   | 10°W–45°E, 38°N–70°N  |  |  |
| East Asia                | 90°E–150°E/10°N–55°N  |  |  |
| South America            | 82°W–30°W/40°S–15°N   |  |  |
| Tropical Africa          | 15°W–55°E/10°S–20°N   |  |  |
| Northern Africa          | 15°W–55°E/20°N–35°N   |  |  |
| Maritime South East Asia | 90°E–150°E/10°S–10°N  |  |  |
| Tropics                  | 23°S–23°N             |  |  |
| Arctic                   | 60°N–90°N             |  |  |
| Antarctica               | 90°S–60°S             |  |  |
| NH mid latitudes         | 30°N-60°N             |  |  |
| SH mid-latitudes         | 60°S–30°S             |  |  |

Table 3 Coordinates of regions